# Accuracy-First Rényi Differential Privacy and Post-Processing Immunity

**Ossi Räisä** [1]   **Antti Koskela** [2]   **Antti Honkela** [3]

## Abstract

The accuracy-first perspective of differential privacy addresses an important shortcoming by allowing a data analyst to adaptively adjust the quantitative privacy bound instead of sticking to a predetermined bound. Existing works on the accuracy-first perspective have neglected an important property of differential privacy known as post-processing immunity, which ensures that an adversary is not able to weaken the privacy guarantee by post-processing. We address this gap by determining which existing definitions in the accuracy-first perspective have post-processing immunity, and which do not. The only definition with post-processing immunity, pure ex-post privacy, lacks useful tools for practical problems, such as an ex-post analogue of the Gaussian mechanism, and an algorithm to check if accuracy on separate private validation set is high enough. To address this, we propose a new definition based on Rényi differential privacy that has post-processing immunity, and we develop basic theory and tools needed for practical applications. We demonstrate the practicality of our theory with applications to synthetic data generation and image classifier fine-tuning, where our algorithm successfully adjusts the privacy bound until an accuracy threshold is met on a private validation dataset.

## 1. Introduction

As the volume of data collection and capabilities of machine learning models increase, the need for methods to use the data while maintaining the privacy of data subjects grows ever larger. Differential privacy (DP; Dwork et al., 2006b) has become the gold-standard definition of privacy

to address this need.

However, differential privacy is a very strict definition. First, it requires protection against a worst-case adversary with very strong capabilities, which necessitates very strong protection that can heavily decrease the utility of an analysis. Second, it requires specifying a numerical privacy bound before running the algorithm, which precludes workflows such as reducing the privacy bound over time in case the data becomes less sensitive, using different bounds for different audiences (Rosenblatt et al., 2024), and addressing low utility by decreasing the privacy protection afterwards (Ligett et al., 2017; Nanayakkara et al., 2024).

Motivated by these issues, Ligett et al. (2017) proposed the *accuracy-first* perspective to DP, where the analyst sets a minimum utility (accuracy) threshold, and finds the strongest privacy bound that empirically meets the threshold. As the privacy bound now depends on the private data, standard DP definitions, known as *ex-ante* definitions, are not applicable. Instead, this perspective requires a new type of privacy definition that allows the privacy bound to depend on the private data, known as *ex-post privacy*. Ligett et al. (2017) proposed the first ex-post privacy definition, pure ex-post privacy, and developed a *Laplace noise reduction* mechanism that satisfies their definition. In subsequent work, Whitehouse et al. (2022) proposed a probabilistic relaxation, and developed the *Brownian mechanism*, an analogue of the Gaussian mechanism (Dwork et al., 2006a) of ex-ante DP providing higher utility. In parallel to our work, Ghazi et al. (2025) developed a hyperparameter tuning algorithm for the accuracy-first perspective, and proposed ex-post privacy definitions based on *approximate DP* (ADP, Dwork et al., 2006a) and *Rényi DP* (RDP; Mironov, 2017).

However, these works have neglected an important aspect that any privacy definition should have, known as *post-processing immunity* (Dwork & Roth, 2014). Post-processing immunity is the requirement that any processing of the output of a private algorithm cannot decrease the privacy guarantee. It is clear why any sensible privacy definition (Mironov, 2017; Dong et al., 2022) should have post-processing immunity: without it, an adversary could use post-processing to infer private information more easily than promised, making the guarantee weaker than promised (Meiser, 2018).

---

[1]CISPA Helmholtz Center for Information Security, Saarbrücken, Germany. Work primarily done at University of Helsinki [2]Nokia Bell Labs, Espoo, Finland [3]University of Helsinki, Department of Computer Science, Helsinki, Finland. Correspondence to: Ossi Räisä <ossi.raisa@cispa.de>.

*Proceedings of the 43rd International Conference on Machine Learning*, Seoul, South Korea. PMLR 306, 2026. Copyright 2026 by the author(s).

**Contributions** Our goal is to address this gap by determining whether previous ex-post privacy definitions have post-processing immunity, and by developing additional theory that supports mechanisms for practical problems under ex-post privacy guarantees with post-processing immunity.

1. We propose a new notation for ex-post privacy definitions that makes the promised privacy bound a part of the output of the algorithm, instead of making it a separate function of the output. The new notation allows defining post-processing immunity in an elegant way, while we were not able to find a satisfactory definition with the old notation, and neither has any previous work (Ligett et al., 2017; Whitehouse et al., 2022; Ghazi et al., 2025) that we know of. We then formulate the existing definitions to use our new notation. See Section 3.1 for details.

2. After the translation to the new notation, we prove that the pure ex-post privacy definition of Ligett et al. (2017) has post-processing immunity, while the probabilistic relaxation of Whitehouse et al. (2022) does not have it. This motivates the development of an alternative relaxation that both has post-processing immunity, and supports the Brownian mechanism. See Section 3.2 for details, and Table 1 for a summary.

3. We investigate an ex-post privacy definition based on RDP, which has been proposed in parallel by Ghazi et al. (2025). We prove that this definition has post-processing immunity, along with many basic properties that are useful in constructing practical mechanisms. In particular, we provide an algorithm that checks whether accuracy on a private validation dataset is high enough, whereas previous works only provide algorithms to check accuracy on the training set (Ligett et al., 2017; Whitehouse et al., 2022). See Section 4.

4. We prove that the Brownian mechanism satisfies ex-post RDP in Section 5. This is the first time the Brownian mechanism has been proven to satisfy an ex-post definition with post-processing immunity. Our proof uses a more elementary view of the Brownian mechanism than Whitehouse et al. (2022), that does not require working with stochastic processes, which may be of interest to future work.

5. We give a practical example of using the Brownian mechanism and our theory in Section 6. We generate synthetic data, and ensure that the accuracy of a classifier trained on the synthetic data on a private validation dataset is high enough on a minimal privacy budget. We find that our implementation is able to achieve high accuracy while keeping a tight privacy budget. We give another example in Appendix E, where we fine-tune an image classifier, ensuring a minimum accuracy is met while minimising the privacy budget.

### 1.1. Related Work

The closest related works are those by Ligett et al. (2017), Whitehouse et al. (2022) and Ghazi et al. (2025) that have proposed several ex-post privacy definitions. Follow-up work (Rogers et al., 2023) has studied compositions involving ex-post private mechanisms. None of these have proven that any ex-post privacy definition has post-processing immunity.

Another line of related work is known as privacy odometers (Rogers et al., 2016). A privacy odometer allows running a sequence of mechanisms with adaptively selected privacy bounds and keeps track of the cumulative privacy budget that has been spent. Odometers have been developed for several DP definitions (Rogers et al., 2016; Feldman & Zrnic, 2021; Lécuyer, 2021; Koskela et al., 2023; Whitehouse et al., 2023) and more general sequences of mechanisms (Haney et al., 2023). While odometers do not provide ex-ante DP guarantees, they can provide ex-post guarantees. See Appendix A for more details.

## 2. Background

In this section, we introduce required background material on DP (Section 2.1) and ex-post privacy (Section 2.2).

**Common Notations** We use $p_P(y)$ to denote the density function of a random variable $P$ at a value $y$. We do not necessarily assume that $P$ is continuous, but rather assume that $P$ has a density function with regards to some base measure $\mu$, which is a much more general condition[1]. This allows us to treat discrete and continuous distributions in the same way.

We use $y_{i:j}$ to denote a sequence of values from index $i$ to $j$. If $j < i$, we take the sequence to be empty. Table 2 in the Appendix contains summary of common notations we use.

### 2.1. Differential Privacy Basics

Differential privacy quantitatively formalises the notion of privacy loss resulting from the release of an output that depends on private input data. This can be formalised in several ways. For the standard ex-ante setting with the privacy bound fixed upfront, we present two definitions. The first is the most commonly used definition, known as *approximate DP* (ADP) or $(\epsilon, \delta)$-DP. The second is known as *Rényi DP* (RDP).

**Definition 2.1** (Dwork et al. 2006a). *An algorithm $\mathcal{M} \colon \mathcal{X} \to \mathcal{Y}$ is $(\epsilon, \delta)$-DP for $\epsilon \geq 0$, $0 \leq \delta < 1$ if, for all neighbouring $X, X' \in \mathcal{X}$ and all measurable $A \subset \mathcal{Y}$,*

$$\Pr(\mathcal{M}(X) \in A) \leq e^\epsilon \Pr(\mathcal{M}(X') \in A) + \delta. \quad (1)$$

---

[1]This means that there exists a base measure $\mu$ such that for any measurable set $A$, $\Pr(A) = \int_A p_P(y) \, d\mu(y)$.

The case with $\delta = 0$ is also known as pure DP, and denoted $\epsilon$-DP.

**Definition 2.2** (Mironov, 2017). *An algorithm $\mathcal{M}\colon \mathcal{X} \to \mathcal{Y}$ is $(\alpha, \epsilon)$-RDP for $\epsilon \geq 0$, $\alpha > 1$ if, for all neighbouring $X, X' \in \mathcal{X}$,*

$$\frac{1}{\alpha - 1} \ln \mathop{\mathbb{E}}_{y \sim \mathcal{M}(X')} \left[ \left( \frac{p_{\mathcal{M}(X)}(y)}{p_{\mathcal{M}(X')}(y)} \right)^{\alpha} \right] \leq \epsilon. \quad (2)$$

It is possible to define RDP for $\alpha = 1$ and $\alpha = \infty$ by taking a limit of the left hand side, but we will not consider those values in this paper.

DP algorithms are also called mechanisms. We use $X \sim X'$ to denote neighbouring datasets. The neighbourhood relation $\sim$ is domain-specific, but the results in this paper, with the exception of the experiments in Section 6 and Appendix E, are agnostic to the definition.

We will also make use of *privacy loss functions* (PLF) and *privacy loss random variables* (PLRV).

**Definition 2.3** (Sommer et al. 2019). *For random variables $P, Q$, with respective density functions $p_P, p_Q$, the PLF is $\mathcal{L}(y) = \ln \frac{p_P(y)}{p_Q(y)}$ and the PLRV is $L = \mathcal{L}(y)$, where $y \sim P$.*

The PLF $\mathcal{L}_{X/X'}^{\mathcal{M}}$ and PLRV $L_{X/X'}^{\mathcal{M}}$ for a mechanism $\mathcal{M}$ and datasets $X \sim X'$ is formed by setting $P = \mathcal{M}(X)$ and $Q = \mathcal{M}(X')$. The definition of $(\epsilon, 0)$-DP is then equivalent to $\Pr(L_{X/X'}^{\mathcal{M}} > \epsilon) = 0$ for all $X \sim X'$.

## 2.2. Accuracy-First Differential Privacy

In the accuracy-first perspective of DP, the privacy bound is not fixed upfront, and can vary based on the result of a mechanism, possibly in a data-dependent way. Since common ex-ante definitions of DP require a fixed privacy bound (Dwork & Roth, 2014; Mironov, 2017; Dong et al., 2022), the accuracy-first perspective requires generalised definitions that can accommodate a data-dependent privacy bound. We call these definitions ex-post privacy definitions after the first such definition from Ligett et al. (2017).

**Definition 2.4** (Ligett et al. 2017; Whitehouse et al. 2022). *An algorithm $\mathcal{M}\colon \mathcal{X} \to \mathcal{Y}$ is $(\mathcal{E}, \delta)$-probabilistically ex-post private for $\mathcal{E}\colon \mathcal{Y} \to \mathbb{R}_{\geq 0}$ if, for all $X \sim X'$,*

$$\Pr_{y \sim \mathcal{M}(X)} (\mathcal{L}_{X/X'}^{\mathcal{M}}(y) > \mathcal{E}(y)) \leq \delta. \quad (3)$$

The original definition from Ligett et al. (2017) has $\delta = 0$. To distinguish it from the $\delta > 0$ case, we will refer to the $\delta = 0$ case as $\mathcal{E}$-pure ex-post privacy. These definitions accommodate a varying privacy bound by making the bound a function $\mathcal{E}$ of the output $y$.

Recently, Ghazi et al. (2025) have proposed[2] an ex-post privacy definition similar to ADP, which we call *approximate ex-post privacy* to distinguish from the previous definition.

**Definition 2.5** (Ghazi et al., 2025). *An algorithm $\mathcal{M}\colon \mathcal{X} \to \mathcal{Y}$ is $(\mathcal{E}, \delta)$-approximately ex-post private for $\mathcal{E}\colon \mathcal{Y} \to \mathbb{R}_{\geq 0}$ if, for all $X \sim X'$ and measurable $S \subseteq \mathcal{Y}$,*

$$\Pr_{y \sim \mathcal{M}(X)} (y \in S) \leq \mathop{\mathbb{E}}_{y \sim \mathcal{M}(X')} [e^{\mathcal{E}(y)} \mathbb{I}_{y \in S}] + \delta. \quad (4)$$

Due to the recency of this definition, we leave closer examinations of it for future work.

## 3. Post-Processing in Ex-Post Privacy

In ex-ante DP, post-processing immunity requires that processing the output of a DP mechanism without further access to the input data cannot weaken the DP guarantee (Dwork & Roth, 2014). Any sensible privacy definition should have post-processing immunity (Mironov, 2017; Dong et al., 2022), since if an adversary can weaken a privacy guarantee by simply post-processing, that guarantee was not real in the first place (Meiser, 2018). Indeed, common ex-ante DP definitions have post-processing immunity (Dwork et al., 2006a;b; Bun & Steinke, 2016; Mironov, 2017; Dong et al., 2022).

In addition, post-processing immunity is extremely useful in constructing DP mechanisms to solve practical problems, as it allows choosing an intermediate quantity to release with a simple DP mechanism, and using that to solve the problem without needing to consider DP further. Many DP algorithms for solving complex problems, such as DP-SGD (Rajkumar & Agarwal, 2012; Song et al., 2013; Abadi et al., 2016), and DP synthetic data generation (Hardt et al., 2012), use this design.

In contrast to ex-ante DP, we are not aware of prior work proving that any ex-post privacy definition has post-processing immunity. To rectify this problem, we start with a very basic issue: the notation used in ex-post privacy definitions (Section 3.1). We then determine whether pure ex-post privacy and $\delta$-probabilistic ex-post privacy have post-processing immunity (Section 3.2). See Table 1 for a summary of the results.

### 3.1. Ex-post Privacy Notation

The original notation for ex-post private mechanisms, presented in Section 2.2, has a major problem when considering post-processing. All of the definitions use a function $\mathcal{E}\colon \mathcal{Y} \to \mathbb{R}_{\geq 0}$ that computes the privacy bound based on the

---

[2]Ghazi et al. (2025) credit this definition to Wu et al. (2019), which is the journal version of Ligett et al. (2017), but we did not find the definition in that paper, and are not aware of any other work proposing it.

*Table 1.* Summary of ex-post privacy definitions. The columns indicate whether the definition has post-processing immunity (PPI) and whether the Brownian mechanism (BM) satisfies it. Whether $\delta$-approximate ex-post privacy has PPI is currently unknown, indicated by "?". The PPI column shows the status for the formulations of the definitions using our notation from Section 3.1.

| Definition | PPI | BM | Source |
|---|---|---|---|
| Pure ex-post privacy | ✓* | × | Ligett et al., 2017, Definition 2.3 |
| $\delta$-probabilistic ex-post privacy | ×* | ✓ | Whitehouse et al., 2022, Definition 2.2 |
| $\delta$-approximate ex-post privacy | ? | ✓* | Ghazi et al., 2025, Definition 5 |
| $\alpha$-ex-post RDP | ✓* | ✓* | This work, Definition 4.1 |

*Proven in this work.

output. For simplicity, in the following we simply refer to $\mathcal{E}$-ex-post privacy, leaving out the $\delta$ and the exact definition, as this issue applies to all of the definitions.

A naive statement of post-processing immunity would say that if a randomised function $f\colon \mathcal{Y} \to \mathcal{Y}'$ is applied to the output of an $\mathcal{E}$-ex-post private mechanism $\mathcal{M}\colon \mathcal{X} \to \mathcal{Y}$, the resulting mechanism $\mathcal{M}' = f \circ \mathcal{M}$ is $\mathcal{E}$-ex-post private. However, this statement is not coherent: the domain of $\mathcal{E}$ must match the range of the mechanism, but the domain of $\mathcal{E}$ is $\mathcal{Y}$, while the range of $\mathcal{M}'$ is $\mathcal{Y}'$. To state post-processing immunity, we would need to construct a function $\mathcal{E}'\colon \mathcal{Y}' \to \mathbb{R}_{\geq 0}$ that is in some way equivalent to $\mathcal{E}$.

However, we have not been able to come up with a satisfactory construction of $\mathcal{E}'$ that works for all $f$. One could attempt to define $\mathcal{E}'(y') = \sup_{y \in f^{-1}(y')} \mathcal{E}(y)$, where $f^{-1}$ denotes the preimage. However, this is potentially extremely pessimistic. For example, consider the Brownian mechanism in the form presented by Whitehouse et al. (2022), where the mechanism releases the output $y \in \mathbb{R}$ and the final noise variance $T > 0$, and $\mathcal{E}$ is computed from $T$. Suppose that the noise variances are calibrated in such a way that $\mathcal{E}(T) \in [0.1, 10]$. Consider the post-processing $g(y, T) = y$ that forgets $T$. Since a particular value of $y$ as output does not rule out any $T > 0$ as a possible output, we have $g^{-1}(y) = \{(y, T) \mid T > 0, 0.1 \leq \mathcal{E}(T) \leq 10\}$, so with this definition of $\mathcal{E}'$, we have $\mathcal{E}'(y) = 10$ for all $y \in \mathbb{R}$. This is extremely pessimistic, so $\mathcal{E}'$ is in fact not equivalent to $\mathcal{E}$.

To solve this issue, we introduce a new notation that makes the privacy bound a part of the output of the mechanism. Mathematically, we define a mechanism in the ex-post setting to be a randomised function $\mathcal{M}\colon \mathcal{X} \to \mathcal{Y} \times \mathbb{R}_{\geq 0}$. We use $\mathcal{M}(X)_y \in \mathcal{Y}$ to denote the output of interest from the mechanism, and denote the returned privacy bound by $\mathcal{M}(X)_\epsilon \in \mathbb{R}_{\geq 0}$. In total, the mechanism returns the pair $\mathcal{M}(X) = (\mathcal{M}(X)_y, \mathcal{M}(X)_\epsilon)$. Since $\epsilon$ is part of the output, an ex-post privacy definition must account for the fact that releasing $\epsilon$ may also cause privacy loss, so the PLF $\mathcal{L}^{\mathcal{M}}_{X/X'}(y, \epsilon)$ is now a function of the pair $(y, \epsilon)$. In the Brownian mechanism example from the previous paragraph,

the mechanism returns $((y, T), \mathcal{E}(T))$ instead of just $(y, T)$ with this notation.

Next, we present a formulation of the probabilistic ex-post privacy definition (Definition 2.4) using our notation. The other definitions can be converted similarly. As the privacy bound is now part of the mechanism's output, we drop the $\mathcal{E}$ from the definition, so it is now called $\delta$-probabilistic ex-post privacy.

**Definition 3.1** ($\delta$-probabilistic ex-post privacy)**.** *An algorithm* $\mathcal{M}\colon \mathcal{X} \to \mathcal{Y} \times \mathbb{R}_{\geq 0}$ *is* $\delta$-*probabilistically ex-post private if, for all* $X \sim X'$,

$$\Pr_{(y,\epsilon) \sim \mathcal{M}(X)} (\mathcal{L}^{\mathcal{M}}_{X/X'}(y, \epsilon) > \epsilon) \leq \delta. \tag{5}$$

Now we can elegantly define post-processing for ex-post mechanisms. Since this is different from function composition, we use $\odot$ to denote this operation.

**Definition 3.2.** *Let* $\mathcal{M}\colon \mathcal{X} \to \mathcal{Y} \times \mathbb{R}_{\geq 0}$ *be a mechanism and let* $f\colon \mathcal{Y} \times \mathbb{R}_{\geq 0} \to \mathcal{Y}'$ *be a randomised function. We define the post-processing of* $\mathcal{M}$ *with* $f$, *denoted* $\mathcal{M}' = f \odot \mathcal{M}$, *to be the mechanism*

$$\mathcal{M}'(X)_y = f\big(\mathcal{M}(X)\big), \tag{6}$$
$$\mathcal{M}'(X)_\epsilon = \mathcal{M}(X)_\epsilon. \tag{7}$$

With this definition of post-processing, we can simply define post-processing immunity to mean that $\mathcal{M}$ and $\mathcal{M}'$ satisfy the same ex-post privacy notion, whether it be $\delta$-probabilistic ex-post privacy or any other definition. In Section 3.2, we show that Definition 3.1 satisfies post-processing immunity when $\delta = 0$, but not when $\delta > 0$.

As $\epsilon$ is part of the mechanism's output, we allow post-processing both the output of interest $y$ and $\epsilon$, but the returned privacy bound of the post-processed mechanism does not change. This is in line with the capabilities of an adversary: they can arbitrarily attempt to find an easier representation of $(y, \epsilon)$ to attack, but they cannot change what the original mechanism claimed the privacy bound to be.

Disregarding post-processing, Definition 3.1 and Definition 2.4 are equivalent in the following sense. First, con-

sider a pair $(\mathcal{M}, \mathcal{E})$ from Definition 2.4. We can define $\mathcal{M}'(X)_y = \mathcal{M}(X)$ and $\mathcal{M}'(X)_\epsilon = \mathcal{E}(\mathcal{M}(X))$. Now $\mathcal{M}'$ satisfies Definition 3.1 if and only if $(\mathcal{M}, \mathcal{E})$ satisfy Definition 2.4. In the other direction, if we have a mechanism $\mathcal{M}$ from Definition 3.1, we can define $\mathcal{M}'(X) = (\mathcal{M}(X)_y, \mathcal{M}(X)_\epsilon)$ and $\mathcal{E}(y, \epsilon) = \epsilon$. Again, $\mathcal{M}'$ satisfies Definition 2.4 if and only if $\mathcal{M}$ satisfies Definition 3.1. This means that making the privacy bound an output of the algorithm in Definition 3.1 is only a conceptual change from Definition 2.4, and does not limit the applicability of the definition.

### 3.2. Post-Processing Immunity Status for Existing Definitions

Next, we show that the original ex-post privacy definition (Ligett et al., 2017) (Definition 3.1 with $\delta = 0$) has post-processing immunity. We defer all proofs in this section to Appendix C.2.

**Theorem 3.3.** *Let $\mathcal{M}$ be a pure ex-post private mechanism with continuous (or discrete) output distributions and let $f$ be a randomised function with continuous (or discrete) output distributions. Then the post-processed mechanism $\mathcal{M}' = f \odot \mathcal{M}$ (see Definition 3.2) is pure ex-post private.*

Note that unlike our other theory, we were only able to prove this theorem for the continuous and discrete special cases, and leave a more general treatment to future work.

In contrast, next we show that $\delta$-probabilistic ex-post privacy with $\delta > 0$ does not have post-processing immunity. This is because $\delta$-probabilistic ex-post privacy generalises $(\epsilon, \delta)$-probabilistic-DP, which we define next.

**Definition 3.4** (Meiser 2018, Definition 4). *A mechanism $\mathcal{M}$ is $(\epsilon, \delta)$-probabilistically DP if, for all neighbouring datasets $X, X'$, there is a set $S^\delta$ with $\Pr(\mathcal{M}(X) \in S^\delta) \leq \delta$ such that for all measurable $S$,*

$$\Pr(\mathcal{M}(X) \in S \setminus S^\delta) \leq e^\epsilon \Pr(\mathcal{M}(X') \in S \setminus S^\delta). \quad (8)$$

Meiser (2018) shows that Definition 3.4 does not have post-processing immunity. The simplest counterexample has a mechanism with binary input and four possible outputs, and a post-processing that simply combines two of the possible outputs into one. The probabilities of each input-output pair are chosen such that the original mechanism is $(\epsilon, \delta)$-probabilistically DP, but the post-processed one is not.

It turns out that Definition 3.4 is a special case of $\delta$-probabilistic ex-post privacy, which means that $\delta$-probabilistic ex-post privacy cannot have post-processing immunity either when $\delta > 0$.

**Theorem 3.5.** *For mechanisms $\mathcal{M}$ with constant $\mathcal{M}(X)_\epsilon = \epsilon_c$, $\delta$-probabilistic ex-post privacy (Definition 3.1) is equivalent to $(\epsilon_c, \delta)$-probabilistic DP (Definition 3.4).*

## 4. Ex-post RDP

In this section, we define ex-post RDP and prove some basic properties. We give a Bayesian interpretation of ex-post RDP similar to that of Mironov (2017) for ex-ante RDP in Appendix B.

**Definition of ex-post RDP** The definition of RDP uses the following inequality:

$$\frac{1}{\alpha - 1} \ln \mathop{\mathbb{E}}_{y \sim \mathcal{M}(X')} \left[ \left( \frac{p_{\mathcal{M}(X)}(y)}{p_{\mathcal{M}(X')}(y)} \right)^\alpha \right] \leq \epsilon. \quad (9)$$

It is possible to move $\epsilon$ inside the expectation and arrive at an equivalent inequality:

$$\mathop{\mathbb{E}}_{y \sim \mathcal{M}(X')} \left[ e^{(1-\alpha)\epsilon} \left( \frac{p_{\mathcal{M}(X)}(y)}{p_{\mathcal{M}(X')}(y)} \right)^\alpha \right] \leq 1. \quad (10)$$

This motivates an ex-post version of RDP:

**Definition 4.1.** *A mechanism $\mathcal{M} \colon \mathcal{X} \to \mathcal{Y} \times \mathbb{R}_{\geq 0}$ is $\alpha$-ex-post RDP ($\alpha > 1$) if, for all neighbouring $X \sim \bar{X}'$,*

$$\mathop{\mathbb{E}}_{(y,\epsilon) \sim \mathcal{M}(X')} \left[ e^{(1-\alpha)\epsilon} \left( \frac{p_{\mathcal{M}(X)}(y, \epsilon)}{p_{\mathcal{M}(X')}(y, \epsilon)} \right)^\alpha \right] \leq 1. \quad (11)$$

Note that unlike the definition of RDP, we do not permit $\alpha = 1$ or $\alpha = \infty$ in this work, and leave the investigation whether ex-post RDP could be extended to them for future work. The reason is that the limits used to define these cases for RDP are not as well-understood when $\epsilon$ is a random variable inside the expectation. This definition has also been proposed in parallel work (Ghazi et al., 2025), though using the original ex-post privacy notation instead of our notation.

The conceptual difference between Definition 4.1 and Definition 3.1 is similar to the difference between probabilistic DP and RDP. Definition 3.1 is essentially a tail bound on the random variable $\mathcal{L}_{X/X'}^{\mathcal{M}}(y, \epsilon)$, while Definition 4.1 essentially bounds the moment-generating function of $\mathcal{L}_{X/X'}^{\mathcal{M}}(y, \epsilon)$. The word "essentially" is needed because the bound $\epsilon$ is also a random variable in the ex-post case.

We also present an ex-post version of zero-concentrated DP (zCDP; Bun & Steinke, 2016) in Appendix D.2 (Definition D.2) that relates to ex-post RDP the same way the ex-ante versions are related.

**Basic Properties of Ex-post RDP** In the rest of this section, we prove many basic theorems about ex-post RDP that we use to construct a practical mechanism in Section 6. We present proof ideas in the main text, and defer full proofs to Appendix C.3. We start with post-processing immunity.

**Theorem 4.2.** *Let $\mathcal{M}$ be an $\alpha$-ex-post RDP mechanism and let $f$ be a randomised function. Then the post-processed mechanism $\mathcal{M}'(X) = f \odot \mathcal{M}$ (Definition 3.2) is $\alpha$-ex-post RDP.*

*Proof idea.* The idea of the proof is to use the data-processing inequality (van Erven & Harremoës, 2014), similar to the ex-ante case (Mironov, 2017). □

Next, we prove that if an ex-post RDP mechanism has an upper bound on the returned $\epsilon$, the mechanism is ex-ante RDP.

**Theorem 4.3.** *Let $\mathcal{M}$ be an $\alpha$-ex-post RDP mechanism with $\mathcal{M}(X)_\epsilon \leq \epsilon_c$ almost surely for all $X \in \mathcal{X}$. Then $\mathcal{M}$ is $(\alpha, \epsilon_c)$-RDP.*

*Proof idea.* The claim follows by plugging $\epsilon_c$ into Definition 4.1 and basic manipulations. □

The next theorem shows that ex-post RDP mechanisms can be adaptively composed. This also covers compositions mixing ex-ante RDP and ex-post RDP mechanism, as ex-ante RDP is a special case of ex-post RDP.

**Theorem 4.4.** *Let $\mathcal{M}_i(X, y_{1:i-1})$, $1 \leq i \leq T$ be $\alpha$-ex-post RDP mechanisms. Then the adaptive composition $\mathcal{M}^*(X) = (y_{1:T}, \sum_{i=1}^{T} \epsilon_i)$ with*

$$y_i = \mathcal{M}_i(X, y_{1:i-1})_y \tag{12}$$
$$\epsilon_i = \mathcal{M}_i(X, y_{1:i-1})_\epsilon \tag{13}$$

*is $\alpha$-ex-post RDP. If $\mathcal{M}^*$ additionally releases the intermediate $\epsilon_i$ values, it remains $\alpha$-ex-post RDP.*

*Proof idea.* The idea of the proof is to use the law of total expectation and the chain rule of probabilities to decompose expectation in Definition 4.1 for $\mathcal{M}^*$, and then use the ex-post RDP guarantees for each $\mathcal{M}_i$ to bound each part of the decomposition. □

The next theorem shows how an ex-post RDP mechanism can be constructed out of a composition of RDP mechanisms with fully adaptive privacy bounds. Note that we have changed the number of mechanisms to $K$. We discuss the reason for this after the theorem.

**Theorem 4.5.** *Let $\mathcal{M}_i(X, \epsilon, y_{1:i-1})$, $1 \leq i \leq K$ be mechanisms that are $(\alpha, \epsilon)$-RDP for any $\epsilon \geq 0$. Then the adaptive composition with adaptive privacy bounds $\mathcal{M}^*(X)$ defined as*

$$y_1 = \mathcal{M}_1(X, \epsilon_1, \emptyset) \tag{14}$$
$$\epsilon_i = \mathcal{E}(y_{1:i-1}, \epsilon_{1:i-1}) \tag{15}$$
$$y_i = \mathcal{M}_i(X, \epsilon_i, y_{1:i-1}), \tag{16}$$

*$\mathcal{M}^*(X)_y = y_{1:K}$, and $\mathcal{M}^*(X)_\epsilon = \sum_{i=1}^{K} \epsilon_i$ is $\alpha$-ex-post RDP. The mechanism that additionally releases the intermediate $\epsilon_{1:K}$ values is also $\alpha$-ex-post RDP.*

*Proof idea.* The idea of the proof is again using the law of total expectation and the chain rule of probabilities to decompose expectation in Definition 4.1 for $\mathcal{M}^*$. The proof then uses the RDP guarantees of each $\mathcal{M}_i$ to bound each part of the decomposition. □

While the theorem assumes that there is a fixed number $K$ of mechanisms that are run, in practice one can stop early based on the released values by setting $\mathcal{E}(y_{1:i-1}, \epsilon_{1:i-1}) = 0$ for all iterations after $i$. This makes $K$ the maximum number of iterations that can be run, but this has no practical effect, since one can set $K$ to be an arbitrarily large number without any downsides. This is why we used $K$ to denote the number of mechanisms in the theorem, in contrast to $T$ in Theorem 4.4.

The previous theorem does not allow the privacy bounds to depend on private data, except through the outputs of the base mechanisms. The following theorem allows a private data-dependent stopping rule, shown in Algorithm 1. Unlike similar results in previous works (Ligett et al., 2017; Whitehouse et al., 2022), this theorem uses a separate private dataset for the stopping criterion, which, in machine learning applications, allows checking whether accuracy crosses a threshold on validation data instead of training data.

---

**Algorithm 1:** Ex-post Composition with Data-Dependent Stopping Criterion

---

**Input:** Dataset $X$ divided into train set $X_y$ and validation set $X_\epsilon$, $(\alpha, \epsilon_i)$-RDP mechanisms $\mathcal{M}_i$, $(\alpha, \epsilon_{\mathcal{T}})$-RDP stopping rule $\mathcal{T}$, privacy bound choosing rule $\mathcal{E}$

**Output:** Number of outputs $t$, sequence of outputs $y_{1:t}$, privacy bounds $\epsilon_{1:t}$, total privacy bound $\epsilon$

**for** $1 \leq i \leq K$ **do**
  $\epsilon_i \leftarrow \mathcal{E}(y_{1:i-1}, \epsilon_{1:i-1})$;
  $y_i \leftarrow \mathcal{M}_i(X_y, \epsilon_i, y_{1:i-1})$;
  $c \leftarrow \mathcal{T}(X_\epsilon, y_i)$;
  **if** $c = \text{Halt}$ **then**
    $t \leftarrow i$;
    $\epsilon \leftarrow \max\left(\sum_{j=1}^{t} \epsilon_j, \epsilon_{\mathcal{T}}\right)$;
    **return** $t, y_{1:t}, \epsilon_{1:t}, \epsilon$;
  **end**
**end**

---

**Theorem 4.6.** *Let a dataset $X$ be divided into a training set $X_y$ and validation set $X_\epsilon$, that are disjoint, in any data-independent manner that ensures any neighbouring datasets $X_1 \sim X_2$ only differ in one part of the split. Let $\mathcal{M}_i(X_y, \epsilon_i, y_{1:i-1})$, $1 \leq i \leq K$ be mechanisms that are $(\alpha, \epsilon_i)$-RDP, and let $\mathcal{T}(X_\epsilon, y_{1:i-1})$ be a stopping rule that*

*is $(\alpha, \epsilon_{\mathcal{T}})$-RDP, $\alpha > 1$. Then the adaptive composition with adaptive data-dependent privacy bounds $\mathcal{M}^*$ defined in Algorithm 1 with $\mathcal{M}^*(X)_y = (t, y_{1:t}, \epsilon_{1:t})$, $\mathcal{M}^*(X)_\epsilon = \epsilon$ is $\alpha$-ex-post RDP.*

*Proof idea.* The idea of the proof is to first decompose the expectation in Definition 4.1 for $\mathcal{M}^*$ into two parts: one for the stopping rule $\mathcal{T}$ and one for the composition of the mechanisms $\mathcal{M}_i$. Since their input datasets are disjoint, one of the parts disappears completely, depending on which part the datapoint that changes between $X$ and $X'$ is. If the part for $\mathcal{T}$ remains, it is bounded by its RDP bound. If the part for the composition remains, it is bounded in the same way as in Theorem 4.5. □

## 5. Ex-Post RDP for the Brownian Mechanism

The Brownian mechanism (Whitehouse et al., 2022) is the accuracy-first analogue of the Gaussian mechanism (Dwork et al., 2006a). The idea of the mechanism is to add Gaussian noise to each release of a function $f\colon \mathcal{X} \to \mathbb{R}^d$ like the Gaussian mechanism, and reduce the variance of the noise if the privacy budget is increased for the next release. The added noises at each release are correlated in such a way that the latest release contains all of the information on the private data $X$ that the previous releases contain, so the privacy costs of multiple releases do not accumulate. The appropriate correlations are obtained from a stochastic process called *Brownian motion*. We defer the mathematical definition to Definition C.3 in the Appendix.

We analyse the Brownian mechanism through an alternative perspective that does not use Brownian motion directly. Consider a mechanism that, given an initial RDP bound $(\alpha, \epsilon_1)$, releases a sensitive value $f(X)$ with the Gaussian mechanism $\hat{s}_1 \sim f(X) + \mathcal{N}(0, \sigma_1^2)$, where $\sigma_1^2 = \frac{\alpha \Delta^2}{2\epsilon_1}$ is selected to satisfy $(\alpha, \epsilon_1)$-RDP. For subsequent releases $2 \le i \le K$, the mechanism receives an RDP budget $\epsilon_i = \mathcal{E}(\hat{s}_{1:i-1}, \epsilon_{1:i-1}) > \epsilon_{i-1}$ with constant $\alpha$, sets $\sigma_i^2 = \frac{\alpha \Delta}{2(\epsilon_i - \epsilon_{i-1})}$ according to the extra RDP budget that became available with the increase[3] to $\epsilon_i$. The mechanism then releases $\tilde{s}_i = f(X) + \mathcal{N}(0, \sigma_i^2)$ with the Gaussian mechanism. Finally, the mechanism releases precision-weighted values $\hat{s}_{1:K}$, with $\hat{s}_1 = \tilde{s}_1$ and

$$\hat{s}_i = \left( \sum_{j=1}^{i} \frac{1}{\sigma_j^2} \right)^{-1} \left( \sum_{j=1}^{i} \frac{1}{\sigma_j^2} \tilde{s}_j \right) \quad (17)$$

for $2 \le i \le K$ as the final output. This weighting is the minimum variance unbiased linear weighting for esti-

mating $f(X)$ given the noisy observations $\tilde{s}_i$ with known variances $\sigma_i^2$ (Covington et al., 2025, Theorem 4.1). We call this mechanism the sequential precision-weighted Gaussian mechanism, and summarise it in pseudocode in Algorithm 3 in the Appendix.

Next, we show that the sequential precision-weighted Gaussian mechanism is ex-post RDP using the theory from Section 4, and that the mechanism is equivalent to the Brownian mechanism. We defer the proofs to Appendix C.4.

**Lemma 5.1.** *The sequential precision-weighted Gaussian mechanism (Algorithm 3) is $\alpha$-ex-post RDP.*

The privacy analysis transfers to the Brownian mechanism, since the two mechanisms are equivalent.

**Theorem 5.2.** *For any dataset $X$, function $f$ with $\ell_2$-sensitivity $\Delta$, RDP order $\alpha$, $\epsilon$ selector $\mathcal{E}$, the output of Algorithm 3 has the same distribution as the Brownian mechanism in Algorithm 4.*

**Corollary 5.3.** *The Brownian mechanism (Algorithm 4 in the Appendix) is $\alpha$-ex-post RDP.*

*Proof.* Follows directly from Lemma 5.1 and Theorem 5.2. □

Corollary 5.3 immediately implies that the Brownian mechanism is also approximately ex-post private (Definition 2.5) due to a conversion formula from Ghazi et al. (2025, Lemma 7). In Figure 1, we compare this bound to the probabilistic ex-post privacy bounds of Whitehouse et al. (2022). Note that this comparison assumes that probabilistic ex-post privacy implies approximate ex-post privacy, which we conjecture to hold since it holds for the ex-ante definitions, but has not been proven rigorously. The comparison shows that all bounds are very similar when their free parameters are optimised, so the post-processing immunity and additional flexibility of our RDP-based bound does not come at the cost of utility. Our RDP bound and the linear bound of Whitehouse et al. (2022) are equal within floating-point error, so we conjecture they are mathematically equal.

## 6. Synthetic Data Generation Experiment

To evaluate how the Brownian mechanism works in practice, we generate synthetic tabular data from summary statistics (McKenna et al., 2019) released by the mechanism. We evaluate the quality of synthetic data with the accuracy of a gradient-boosting classifier, and check whether the accuracy meets a minimum threshold on private validation data that is disjoint from training data, applying Theorem 4.6. We use the UCI Adult dataset (Kohavi & Becker, 1996). See Appendix D for details. Code is available at `https://github.com/TrustworthyMLHelsinki/accuracy-first-rdp`.

---

[3]We can handle $\epsilon_i = \epsilon_{i-1}$ by setting $\sigma_i^2 = \infty$ and then using a convention that $\frac{1}{\infty} = 0$ and that a Gaussian distribution with infinite variance has some arbitrary value, which does not matter since it will later always be multiplied by 0.

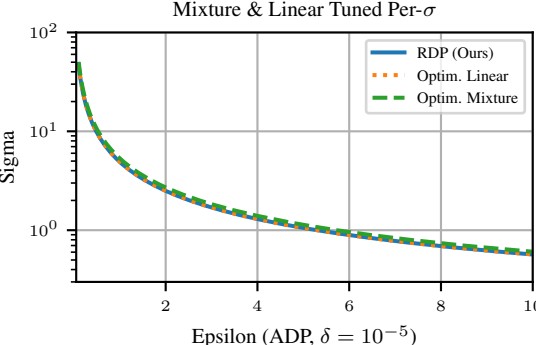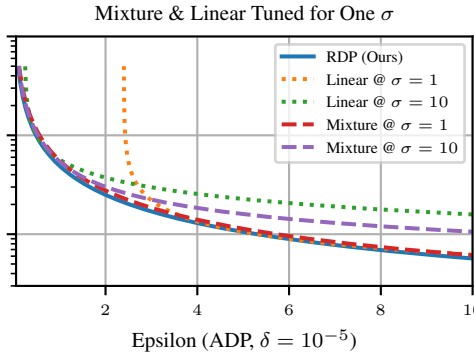

Figure 1. Comparison of our ex-post RDP-based bound and the bounds of Whitehouse et al. (2022) (mixture and linear) for the Brownian mechanism in approximate-ex-post privacy (Definition 2.5), showing all bounds are very similar. This is assuming probabilistic ex-post privacy implies approximate ex-post privacy, which we conjecture is true since it is true in ex-ante DP. In the left panel, we optimise all free parameters of each bound individually for each value of $\sigma$. In the right panel, we use the recommended procedure of Whitehouse et al. (2022) for their bounds, meaning we optimise them for a specific $\sigma$, either 1 or 10. This makes these bounds suboptimal for other values of $\sigma$.

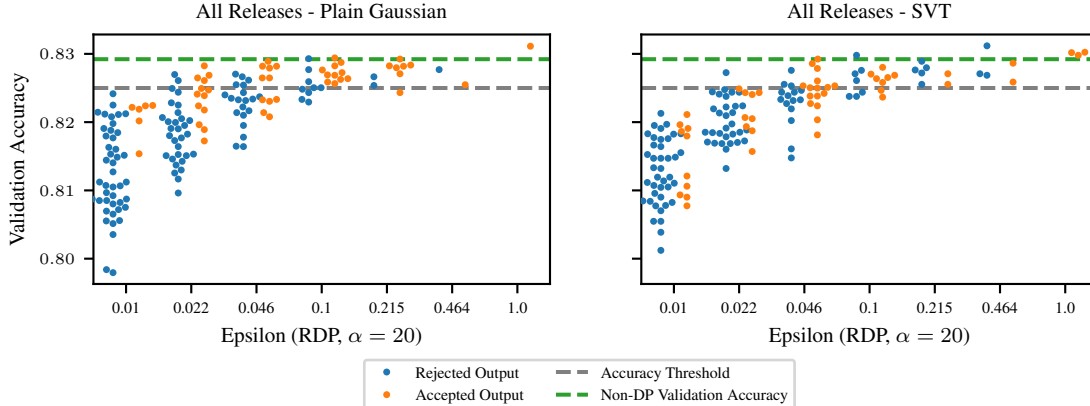

Figure 2. Accuracies and $\epsilon$ values for synthetic data generation on Adult data, showing that the mechanism is able to minimise the privacy cost while ensuring high accuracy. The plain Gaussian (left) mechanism generally accepts higher accuracies than SVT (right) The plot shows results from 50 repeats. The x-axis only shows $\epsilon$ for the synthetic data generation and accuracy threshold checking. The values are logarithmically spaced. For query selection, $\epsilon_{query} = 0.01$, and $\epsilon_{total} = \epsilon_{query} + \epsilon$. Non-DP validation accuracy is computed from synthetic data without DP noise. The plotted value is mean accuracy over 10 repeats of synthetic data generation.

We choose $\epsilon$ values for each release of the Brownian mechanism from a fixed list, containing 7 values, logarithmically spaced between 0.01 and 1. We start by releasing with the lowest $\epsilon$, and if the generated synthetic data does not meet the accuracy threshold, we release with the next $\epsilon$ in the list. This is repeated until the threshold is met, or the list is exhausted. We use the add/remove neighbourhood, where datasets are neighbouring if they are identical except one has an additional datapoint.

We use 20% of the full dataset as test data, and split the rest evenly between training and validation data. Since the validation data we use for the accuracy threshold check is also private, we consider two options to implement the check under DP: a composition of plain Gaussian mechanisms that release noisy accuracies under DP, and sparse vector technique (SVT; Dwork et al., 2009, Zhu & Wang, 2020). Note that both of these checks are noisy, meaning that they

do not always return the correct result of the check. We investigate how changing the size of the validation set affects the results in Appendix D.1, and find that reducing the size from the 40% can improve test accuracy at the cost of a higher final $\epsilon$.

For privacy bounds, we use $\alpha = 20$ throughout. We discuss the effect of changing $\alpha$ in Appendix D.2. We use an RDP preprocessing step (McKenna et al., 2021) to select the summary statistic queries with $\epsilon_{query} = 0.01$, and set $\epsilon_{check} = 0.01$ for checking the accuracy threshold. The $\epsilon$ for synthetic data generation comes from the Brownian mechanism (Corollary 5.3). By Theorems 4.4 and 4.6 and the fact that $\epsilon \geq \epsilon_{check}$, the total privacy bound is $\epsilon_{total} = \epsilon_{query} + \max(\epsilon, \epsilon_{check}) = \epsilon_{query} + \epsilon$. As we focus on the Brownian mechanism, we only include $\epsilon$, not $\epsilon_{total}$, in our plots. Table 3 in the Appendix converts the $\epsilon$ values we use to approximate ex-post privacy (Definition 2.5).

Figure 2 shows the validation accuracies and $\epsilon$ values from 50 repeated runs of the Brownian mechanism and synthetic data generation, with a fixed set of queries. Figure 3 in Appendix D shows distributions of $\epsilon$ values, and validation and test accuracies for the accepted outputs of the mechanism. We see that the Brownian mechanism is able to minimise the privacy cost while approximately meeting the accuracy threshold. The plain Gaussian threshold check generally gives higher accuracies, but SVT gives smaller $\epsilon$ values.

## 7. Discussion and Conclusion

**Interpreting Ex-Post Privacy Bounds**   Since allowing a data-dependent privacy bound in ex-post privacy is a major change from ex-ante privacy, it is important to discuss how ex-post privacy bounds should be interpreted. This applies equally to all ex-post privacy definitions, including ones not using our notation from Section 3.1.

For one run of an ex-post mechanism, the returned privacy bound should be reported, along with the details of how it is computed. The value can then be interpreted in a similar way as the corresponding ex-ante privacy bound, see Appendix B for ex-post RDP and Ligett et al. (2017) for pure ex-post privacy. Values from different runs can easily be compared: the smaller privacy bound is more private.

The more difficult case is comparing mechanisms, in which case the privacy bound is a random variable. A major difficulty is that this random variable depends on the private data, so comparisons using the private data may leak privacy. To circumvent this, it may be possible to use a prior for the private data, and compare the distributions of the privacy bounds under that prior. The distributions can be compared for example by examining their CDFs, though it will not be possible to always establish that one mechanism is more private than the other. We leave an in-depth investigation of this to future work.

**Limitations**   Ex-post RDP inherits many of the limitations of RDP. First, RDP is not as easy to interpret as $f$-DP or Gaussian DP (GDP; Dong et al., 2022), which are defined in terms of optimal performance of a distinguisher. RDP has a Bayesian interpretation (Mironov, 2017), which we extend to ex-post RDP in Appendix B, but these are not as clear as $f$-DP or GDP. Second, the conversion from RDP to approximate DP or $f$-DP is lossy in the ex-ante setting (Zhu et al., 2022), which we expect to also hold in the ex-post setting. For both reasons, we expect developing ex-post GDP or $f$-DP, and determining further properties of approximate ex-post privacy, to be a promising directions for future research.

**Conclusion**   In this work, we determined which ex-post privacy definitions have post-processing immunity and which do not. As the one definition with post-processing immu-

nity does not support the Brownian mechanism, we filled this gap by proposing a new ex-post privacy definition with post-processing immunity, the Brownian mechanism, and supporting theory that enables application to practical problems. We demonstrated two such applications: tabular synthetic data generation and image classifier fine-tuning (Appendix E). In both applications, our mechanism successfully met an accuracy threshold while minimising the spent privacy budget.

## Acknowledgements

This work was supported by the Research Council of Finland (Flagship programme: Finnish Center for Artificial Intelligence, FCAI, Grant 356499 and Grant 359111), the Strategic Research Council at the Research Council of Finland (Grant 358247) as well as the European Union (Project 101070617). Views and opinions expressed are however those of the author(s) only and do not necessarily reflect those of the European Union or the European Commission. Neither the European Union nor the granting authority can be held responsible for them. The authors acknowledge the research environment provided by ELLIS Institute Finland.

## Impact Statement

This work presents theory and practical techniques that aim to improve the privacy of data analyses. While our work, and accuracy-first DP in general, study principled ways to reduce a privacy guarantee, this does not imply that results in this line of work would be used to reduce privacy guarantees in existing analyses. Rather, we think this line of work enables privacy-preserving analyses that were not possible before, and overall has positive societal impact.

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

## Contents

*Table 2.* Common notations.

| | |
|---|---|
| $\mathcal{M}$ | DP Mechanism |
| $\gamma$ | Generic privacy parameter(s) |
| $(\epsilon, \delta)$ | Privacy parameters for ADP-style definitions |
| $(\epsilon, \alpha)$ | Privacy parameters for RDP-style definitions |
| $X \in \mathcal{X}$ | Dataset, universe of datasets |
| $X \sim X'$ | Neighbouring datasets |
| $y \in \mathcal{Y}$ | Output of mechanism, universe of possible outputs |
| $y_{i:j}$ | Sequence from index $i$ to $j$, empty if $j < i$ |
| $\mathcal{M}(X)_y, \mathcal{M}(X)_\epsilon$ | Output and claimed privacy bound of ex-post mechanism $\mathcal{M}$ |
| $P, Q$ | Generic random variables |
| $p_P, p_Q$ | Density functions of $P, Q$, with respect to some base measure |
| $\mathcal{L}(y) = \ln \frac{p_P(y)}{p_Q(y)}$ | Privacy loss function (PLF) of $P, Q$ |
| $L = \mathcal{L}(y), y \sim P$ | Privacy loss random variable (PLRV) of $P, Q$ |
| $\mathcal{L}_{X/X'}^{\mathcal{M}}(y)$ | PLF of ex-ante mechanism $\mathcal{M}$ with regards to $X, X'$ |
| $\mathcal{L}_{X/X'}^{\mathcal{M}}(y, \epsilon)$ | PLF of ex-post mechanism $\mathcal{M}$ with regards to $X, X'$ |
| $L_{X/X'}^{\mathcal{M}}$ | PLRV of mechanism $\mathcal{M}$ with regards to $X, X'$ |

USE OF LARGE LANGUAGE MODELS

Github copilot was used to assist writing the code for the experiments.

## A. Privacy Odometers and Ex-Post Privacy

A closely related line of work to ex-post privacy is the so-called fully adaptive composition that studies *privacy filters* and *odometers* (Rogers et al., 2016; Feldman & Zrnic, 2021; Lécuyer, 2021; Koskela et al., 2023; Whitehouse et al., 2023; Haney et al., 2023). While ex-ante DP composition theorems require specifying the privacy bounds at each step in advance, fully adaptive composition considers the case where the privacy bounds can also be chosen adaptively, based on results from previous iterations. As filters and odometers have been developed for many DP definitions, we use $\gamma \in \Gamma$ to denote a generic privacy bound when discussing filters and odometers in general.

Privacy filters require specifying an upper bound $\gamma_c$ on the privacy loss. The filter keeps track of the accrued privacy loss during the iterations, and halts the computation before the accrued privacy loss exceeds the upper bound. The filter guarantees ex-ante DP with the bound $\gamma_c$.

Privacy odometers do not require specifying an upper bound beforehand. Instead, an odometer keeps track of the accumulated privacy loss, and returns the value of the accumulator at the end of the computation. Interpreting the value returned by the odometer is not as simple as interpreting a filter. While a filter guarantees an ex-ante definition of DP with parameter $\gamma_c$, an odometer does not do so.

We give two reasons why an odometer cannot provide an ex-ante DP guarantee.[4] The first is that the value $\gamma(X)$ returned by the odometer can depend on the data $X$. However, ex-ante definitions of DP do not permit data-dependent privacy bounds. The second reason is more concrete. Consider a pathological mechanism $\mathcal{M}_P$ that releases the input $X$ with probability $\frac{1}{2}$, and releases nothing otherwise. It is well-known that $\mathcal{M}_P$ is not $(\epsilon, \delta)$-DP, or $(\epsilon, \delta)$-probabilistically DP if $\delta < \frac{1}{2}$, and that it is not $(\alpha, \epsilon)$-RDP, unless $\epsilon = \infty$ in either case. However, we can analyse this mechanism with a 1-step odometer, which can return $\epsilon = 0$ for a probabilistic-DP odometer (Rogers et al., 2016) regardless of $\delta$, and return an arbitrarily small $\epsilon$ for an RDP odometer (Feldman & Zrnic, 2021; Lécuyer, 2021). Since we know this mechanism is not ex-ante DP with a finite $\epsilon$ (unless $\delta \geq \frac{1}{2}$), the value returned by the odometer cannot be an ex-ante DP guarantee.

Turning to ex-post DP, we can trivially make the pathological mechanism $\mathcal{M}_P$ ex-post private: return $\epsilon = 0$ if no data is released, and return $\epsilon = \infty$ if data is released. This clearly satisfies $\delta$-probabilistic ex-post privacy (Definition 3.1) even for

---

[4]Note that Lécuyer (2021, Theorem 2) claims that their odometer provides $(\epsilon, \delta)$-DP "for the interaction described in Algorithm 2" (of their paper). Our arguments here show that this is false if it is indeed a claim of ex-ante $(\epsilon, \delta)$-DP. The paper of Lécuyer (2021) does not have any proof of this claim, only the other claim of their Theorem 2 that provides a conversion to a probabilistic DP odometer.

$\delta = 0$, and also satisfies $\alpha$-ex-post RDP (Definition 4.1) for any $\alpha > 1$ if we take $e^{-\infty} = 0$ and $0 \cdot \infty = 0$ in the definition. This begs the question: could privacy odometers provide an ex-post privacy guarantee?

It turns out that odometers can provide an ex-post privacy guarantee. The following theorem shows that a mechanism $\mathcal{M}^*$ computing $\mathcal{M}^*(X)_\epsilon$ with a probabilistic DP odometer (Rogers et al., 2016; Whitehouse et al., 2023) is probabilistically ex-post private.

**Theorem A.1.** *Let* $\mathrm{COMP}_{\delta_g}$ *be a valid privacy odometer in the sense of Rogers et al. (2023, Definition 3.1). Let* $\mathcal{M}^*$ *be an ex-post mechanism where* $\mathcal{M}^*(X)_y$ *runs any fully adaptive composition (Rogers et al., 2023, Algorithm 2), resulting in ADP parameters* $\epsilon_1, \delta_1, \ldots, \epsilon_K, \delta_K$, *and* $\mathcal{M}^*(X)_\epsilon = \mathrm{COMP}_{\delta_g}(\epsilon_1, \delta_1, \ldots, \epsilon_K, \delta_K)$. *Then* $\mathcal{M}^*$ *is* $\delta_g$-*probabilistically ex-post private (Definition 3.1).*

*Proof.* Follows immediately from the definition of validity of $\mathrm{COMP}_{\delta_g}$ according to Rogers et al. (2023, Definition 3.1). □

Combining Theorem A.1 with Theorem 3.3 of Rogers et al. (2016) gives a composition theorem in the style of Theorem 4.5 for pure and probabilistic ex-post privacy.

The ex-post privacy of RDP odometers (Feldman & Zrnic, 2021; Lécuyer, 2021) is not as clear. They are constructed by running multiple privacy filters with different upper bounds, and setting the accumulated privacy loss to be the smallest loss of the filters whose upper bound has not been exceeded. This construction results in some discretisation error in the odometers, since there is only a finite number of filters. Lécuyer (2021) provides a conversion of their odometer to a valid probabilistic-DP odometer in the sense of Rogers et al. (2016), so Theorem A.1 can be applied to it. We have not been able to determine whether either of these odometers implies ex-post RDP (Definition 4.1). However, Theorem 4.5 supersedes an answer to this question, since it essentially provides an odometer that guarantees ex-post RDP without any discretisation error.

Ex-post privacy for the Gaussian DP (GDP; Dong et al., 2022) odometer of Koskela et al. (2023) is even less clear. It is constructed in a similar way to the RDP odometers by running multiple GDP filters. The natural questions to ask is: does this odometer satisfy some sense of ex-post GDP? However, we are not aware of any work giving a definition of ex-post GDP, so this question cannot be answered at this time.

# B. Interpretation of ex-post RDP

Mironov (2017) gives a Bayesian interpretation to RDP based on Bayes factors. Let $R_I(X, X')$ be the Bayes factor between datasets $X \sim X'$ of an adversary with information $I$. That is,

$$R_I(X, X') = \frac{p(X|I)}{p(X'|I)}. \tag{18}$$

Prior to the release of the mechanism's output $y$, the adversary has a prior $R_{\mathrm{prior}}$ with $I = I_{\mathrm{Prior}}$, where $I_{\mathrm{Prior}}$ is whatever prior information the adversary has, and after the release, they have a posterior $R_{\mathrm{post}}$ with $I = \{y\} \cup I_{\mathrm{Prior}}$. $(\alpha, \epsilon)$-RDP bounds the $\alpha - 1$ moment of the change in $R$ as follows (Mironov, 2017):

$$\mathop{\mathbb{E}}_{y \sim \mathcal{M}(X)} \left[ \left( \frac{R_{\mathrm{post}}(X, X')}{R_{\mathrm{prior}}(X, X')} \right)^{\alpha - 1} \right] \leq e^{(\alpha - 1)\epsilon}. \tag{19}$$

We can rearrange (19) to

$$\ln \mathop{\mathbb{E}}_{y \sim \mathcal{M}(X)} \left[ e^{(1 - \alpha)\epsilon} \left( \frac{R_{\mathrm{post}}(X, X')}{R_{\mathrm{prior}}(X, X')} \right)^{\alpha - 1} \right] \leq 0. \tag{20}$$

We can interpret the left hand side as privacy leakage at the $\alpha - 1$ moment exceeding $\epsilon$. Under this interpretation, RDP simply guarantees that there is no excess leakage for a given moment.

Since $\epsilon$ is inside the expectation in this form, we can extend this interpretation to ex-post RDP. We can interpret

$$\ln \mathop{\mathbb{E}}_{y, \epsilon \sim \mathcal{M}(X)} \left[ e^{(1 - \alpha)\epsilon} \left( \frac{R_{\mathrm{post}}(X, X')}{R_{\mathrm{prior}}(X, X')} \right)^{\alpha - 1} \right] \tag{21}$$

as the privacy loss at the $\alpha-1$ moment exceeding the $\epsilon$ the mechanism returned. Note that here $R_{\text{post}}$ has $I = \{(y, \epsilon)\} \cup I_{\text{Prior}}$. Analogously to RDP, ex-post RDP guarantees that there is no excess privacy loss at a given moment.

**Theorem B.1.** *Let $\mathcal{M}$ be an $\alpha$-ex-post RDP mechanism. Then*

$$\ln \mathbb{E}_{y, \epsilon \sim \mathcal{M}(X)} \left[ e^{(1-\alpha)\epsilon} \left( \frac{R_{\text{post}}(X, X')}{R_{\text{prior}}(X, X')} \right)^{\alpha-1} \right] \leq 0. \tag{22}$$

*Proof.* First, by Bayes' theorem,

$$\frac{R_{\text{post}}(X, X')}{R_{\text{prior}}(X, X')} = \frac{p_{\mathcal{M}(X)}(y, \epsilon)}{p_{\mathcal{M}(X')}(y, \epsilon)}. \tag{23}$$

In the expectation in Definition 4.1, we first change the dataset the expectation is taken with respect to, and use the previous to obtain

$$\mathbb{E}_{y, \epsilon \sim \mathcal{M}(X')} \left[ e^{(1-\alpha)\epsilon} \left( \frac{p_{\mathcal{M}(X)}(y, \epsilon)}{p_{\mathcal{M}(X')}(y, \epsilon)} \right)^{\alpha} \right] = \mathbb{E}_{y, \epsilon \sim \mathcal{M}(X)} \left[ e^{(1-\alpha)\epsilon} \left( \frac{p_{\mathcal{M}(X)}(y, \epsilon)}{p_{\mathcal{M}(X')}(y, \epsilon)} \right)^{\alpha-1} \right]$$

$$= \mathbb{E}_{y, \epsilon \sim \mathcal{M}(X)} \left[ e^{(1-\alpha)\epsilon} \left( \frac{R_{\text{post}}(X, X')}{R_{\text{prior}}(X, X')} \right)^{\alpha-1} \right]. \tag{24}$$

Now taking the logarithm of both sides of the inequality in Definition 4.1 proves the claim. $\qquad \square$

## C. Missing Proofs

### C.1. Preliminary Lemmas and Definitions

**Definition C.1** (Rényi divergence; van Erven & Harremoës 2014)**.** *Let $P$ and $Q$ be random variables with respective density functions $p_P$ and $p_Q$. The Rényi divergence of order $\alpha > 1$ between $P$ and $Q$ is*

$$D_\alpha(P \mid\mid Q) = \frac{1}{\alpha - 1} \ln \mathbb{E}_{t \sim Q} \left[ \left( \frac{p_P(t)}{p_Q(t)} \right)^{\alpha} \right] \tag{25}$$

*if $P$ is absolutely continuous with respect to $Q$, and $D_\alpha(P \mid\mid Q) = \infty$ otherwise.*

**Theorem C.2** (Data-processing inequality; van Erven & Harremoës, 2014)**.** *Let $P \in \mathcal{Y}_1$ and $Q \in \mathcal{Y}_1$ be random variables with respective density functions $p_P$ and $p_Q$, and let $f \colon \mathcal{Y}_1 \to \mathcal{Y}_2$ be a randomised function. Let $P' = f \circ P$ and $Q' = f \circ Q$. Then $D_\alpha(P' \mid\mid Q') \leq D_\alpha(P \mid\mid Q)$.*

### C.2. Post-processing Immunity of Existing Definitions

**Theorem 3.3.** *Let $\mathcal{M}$ be a pure ex-post private mechanism with continuous (or discrete) output distributions and let $f$ be a randomised function with continuous (or discrete) output distributions. Then the post-processed mechanism $\mathcal{M}' = f \odot \mathcal{M}$ (see Definition 3.2) is pure ex-post private.*

*Proof.* Let the density functions of $\mathcal{M}(X)$ and $\mathcal{M}(X')$ be $p_{\mathcal{M}(X)}$ and $p_{\mathcal{M}(X')}$. Similarly, let the density functions of $\mathcal{M}'(X)$ and $\mathcal{M}'(X')$ be $p_{\mathcal{M}'(X)}$ and $p_{\mathcal{M}'(X')}$, and let the density of $f(y, \epsilon)$ be $p_{f(y,\epsilon)}$.

We see that the inequality $\mathcal{L}_{X/X'}^{\mathcal{M}}(y, \epsilon) \leq \epsilon$ is equivalent to

$$p_{\mathcal{M}(X)}(y, \epsilon) - e^{\epsilon} p_{\mathcal{M}(X')}(y, \epsilon) \leq 0. \tag{26}$$

If (26) holds, we have:

$$p_{\mathcal{M}'(X)}(y', \epsilon) - e^\epsilon p_{\mathcal{M}'(X')}(y', \epsilon) \tag{27}$$

$$= \int p_{\mathcal{M}(X)}(y, \epsilon) p_{f(y,\epsilon)}(y') \, \mathrm{d}y - e^\epsilon \int p_{\mathcal{M}(X')}(y, \epsilon) p_{f(y,\epsilon)}(y') \, \mathrm{d}y \tag{28}$$

$$= \int p_{f(y,\epsilon)}(y') \left[ p_{\mathcal{M}(X)}(y, \epsilon) - e^\epsilon p_{\mathcal{M}(X')}(y, \epsilon) \right] \mathrm{d}y \tag{29}$$

$$\leq 0, \tag{30}$$

which implies that $\mathcal{L}_{X/X'}^{\mathcal{M}'}(y', \epsilon) \leq \epsilon$. Thus, if $\mathcal{L}_{X/X'}^{\mathcal{M}}(y, \epsilon) \leq \epsilon$ almost surely, then also $\mathcal{L}_{X/X'}^{\mathcal{M}'}(y, \epsilon) \leq \epsilon$ almost surely.

The discrete case follows from a similar argument, replacing density functions with probability mass functions and integrals with sums. $\qquad\square$

**Theorem 3.5.** *For mechanisms $\mathcal{M}$ with constant $\mathcal{M}(X)_\epsilon = \epsilon_c$, $\delta$-probabilistic ex-post privacy (Definition 3.1) is equivalent to $(\epsilon_c, \delta)$-probabilistic DP (Definition 3.4).*

*Proof.* Let the mechanism in question be $\mathcal{M}$, let $X$, $X'$ be neighbouring datasets, and let $p_{\mathcal{M}(X)}$ and $p_{\mathcal{M}(X')}$ be the densities of $\mathcal{M}(X)$ and $\mathcal{M}(X')$ with regards to some measure $\mu$. Even though $\mathcal{M}(X)_\epsilon$ is a constant here, we will formally treat it as a random variable, in line with definition Definition 3.1.

**Definition 3.1 $\Rightarrow$ Definition 3.4** If $\mathcal{M}$ is $\delta$-probabilistically ex-post private with deterministic $\mathcal{M}(X)_\epsilon = \epsilon_c$, we have

$$\Pr_{(y,\epsilon) \sim \mathcal{M}(X)} \left( \ln \frac{p_{\mathcal{M}(X)}(y, \epsilon)}{p_{\mathcal{M}(X')}(y, \epsilon)} > \epsilon \right) \leq \delta, \tag{31}$$

which means

$$\Pr_{(y,\epsilon) \sim \mathcal{M}(X)} \left( p_{\mathcal{M}(X)}(y, \epsilon) > e^\epsilon p_{\mathcal{M}(X')}(y, \epsilon) \right) \leq \delta. \tag{32}$$

Let $S^\delta = \{(y, \epsilon) \mid p_{\mathcal{M}(X)}(y, \epsilon) > e^\epsilon p_{\mathcal{M}(X')}(y, \epsilon)\}$, so $\Pr_{(y,\epsilon) \sim \mathcal{M}(X)}((y, \epsilon) \in S^\delta) \leq \delta$. Now

$$\begin{aligned} \Pr(\mathcal{M}(X) \in S \setminus S^\delta) &= \int_{S \setminus S^\delta} p_{\mathcal{M}(X)}(y, \epsilon) \, \mathrm{d}\mu(y, \epsilon) \\ &\leq \int_{S \setminus S^\delta} e^\epsilon p_{\mathcal{M}(X')}(y, \epsilon) \, \mathrm{d}\mu(y, \epsilon). \end{aligned} \tag{33}$$

Since $\epsilon = \mathcal{M}(X')_\epsilon$ is a deterministic constant, integrating $e^\epsilon \mathbb{I}_{S \setminus S^\delta}$ over the law of $\mathcal{M}(X')$ in the last integral corresponds to a point evaluation with $\epsilon = \epsilon_c$, so

$$\begin{aligned} \int_{S \setminus S^\delta} e^\epsilon p_{\mathcal{M}(X')}(y, \epsilon) \, \mathrm{d}\mu(y, \epsilon) &= e^{\epsilon_c} \int_{S \setminus S^\delta} p_{\mathcal{M}(X')}(y, \epsilon) \, \mathrm{d}\mu(y, \epsilon) \\ &= e^{\epsilon_c} \Pr(\mathcal{M}(X') \in S \setminus S^\delta). \end{aligned} \tag{34}$$

It follows that $\mathcal{M}$ is $(\epsilon_c, \delta)$-probabilistically DP.

**Definition 3.4 $\Rightarrow$ Definition 3.1** If $\mathcal{M}$ is $(\epsilon_c, \delta)$-probabilistically DP, there is a set $S^\delta$ with $\Pr_{(y,\epsilon) \sim \mathcal{M}(X)}((y, \epsilon) \in S^\delta) \leq \delta$ such that the following holds for all measurable $S$:

$$\Pr(\mathcal{M}(X) \in S \setminus S^\delta) \leq e^{\epsilon_c} \Pr(\mathcal{M}(X') \in S \setminus S^\delta). \tag{35}$$

Next, we show that the set

$$S^0 = \{(y, \epsilon) \mid p_{\mathcal{M}(X)}(y, \epsilon) > e^{\epsilon_c} p_{\mathcal{M}(X')}(y, \epsilon)\} \tag{36}$$

has zero base measure outside $S^\delta$, that is, $\mu(S^0 \setminus S^\delta) = 0$. By (35), we have

$$\int_{S^0 \setminus S^\delta} p_{\mathcal{M}(X)}(y, \epsilon) - e^{\epsilon_c} p_{\mathcal{M}(X')}(y, \epsilon) \, \mathrm{d}\mu(y, \epsilon) \leq 0. \tag{37}$$

However, in $S^0 \setminus S^\delta$, $p_{\mathcal{M}(X)}(y, \epsilon) - e^{\epsilon_c} p_{\mathcal{M}(X')}(y, \epsilon) > 0$. Since the integral of a positive function can be non-positive only when integrating over a measure 0 set, we must have $\mu(S^0 \setminus S^\delta) = 0$. This implies that $\Pr_{(y,\epsilon) \sim \mathcal{M}(X)}((y, \epsilon) \in S^0 \setminus S^\delta) = 0$ since $\mu$ is the base measure for $\mathcal{M}(X)$.

Since $p_{\mathcal{M}(X)}(y, \epsilon) \leq e^\epsilon p_{\mathcal{M}(X')}(y, \epsilon)$ when $(y, \epsilon) \notin S^0$, $\mathcal{L}_{X/X'}^{\mathcal{M}}(y, \epsilon) = \ln \frac{p_{\mathcal{M}(X)}(y,\epsilon)}{p_{\mathcal{M}(X')}(y,\epsilon)} \leq \epsilon_c$ when $(y, \epsilon) \notin S^0$. Now

$$
\begin{aligned}
\Pr_{(y,\epsilon) \sim \mathcal{M}(X)} \left( \mathcal{L}_{X/X'}^{\mathcal{M}}(y, \epsilon) > \epsilon \right) &= \Pr_{(y,\epsilon) \sim \mathcal{M}(X)} \left( \mathcal{L}_{X/X'}^{\mathcal{M}}(y, \epsilon) > \epsilon_c \right) \\
&\leq \Pr_{(y,\epsilon) \sim \mathcal{M}(X)} \left( (y, \epsilon) \in S^0 \right) \\
&\leq \Pr_{(y,\epsilon) \sim \mathcal{M}(X)} \left( (y, \epsilon) \in S^\delta \right) + \Pr_{(y,\epsilon) \sim \mathcal{M}(X)} \left( (y, \epsilon) \in S^0 \setminus S^\delta \right) \\
&\leq \delta,
\end{aligned}
\tag{38}
$$

so $\mathcal{M}$ is $\delta$-probabilistically ex-post private. $\qquad\square$

## C.3. Basic ex-post RDP Theorems

**Theorem 4.2.** *Let $\mathcal{M}$ be an $\alpha$-ex-post RDP mechanism and let $f$ be a randomised function. Then the post-processed mechanism $\mathcal{M}'(X) = f \odot \mathcal{M}$ (Definition 3.2) is $\alpha$-ex-post RDP.*

*Proof.* Denote $\beta = e^{(1-\alpha)\epsilon}$. Using the properties of expectations, we have

$$
\begin{aligned}
\mathbb{E}_{(y',\epsilon) \sim \mathcal{M}'(X')} \left[ \beta \left( \frac{p_{\mathcal{M}'(X)}(y', \epsilon)}{p_{\mathcal{M}'(X')}(y', \epsilon)} \right)^\alpha \right] &= \mathbb{E}_\epsilon \mathbb{E}_{y'|\epsilon} \left[ \beta \left( \frac{p_{\mathcal{M}'(X)}(y', \epsilon)}{p_{\mathcal{M}'(X')}(y', \epsilon)} \right)^\alpha \right] \\
&= \mathbb{E}_\epsilon \left[ \beta \left( \frac{p_{\mathcal{M}(X)}(\epsilon)}{p_{\mathcal{M}(X')}(\epsilon)} \right)^\alpha \mathbb{E}_{y'|\epsilon} \left[ \left( \frac{p_{\mathcal{M}'(X)}(y'|\epsilon)}{p_{\mathcal{M}'(X')}(y'|\epsilon)} \right)^\alpha \right] \right] \\
&= (*)
\end{aligned}
\tag{39}
$$

Next, we use the data-processing inequality, Theorem C.2, to bound the inner expectation. Specifically, we set

$$
\begin{aligned}
P &= \mathcal{M}(X)_y | \epsilon, \tag{40} \\
Q &= \mathcal{M}(X')_y | \epsilon, \tag{41} \\
f_{\text{Theorem}}(y) &= f(y, \epsilon), \tag{42}
\end{aligned}
$$

which means that

$$
\begin{aligned}
P' &= \mathcal{M}'(X)_y | \epsilon, \tag{43} \\
Q' &= \mathcal{M}'(X')_y | \epsilon \tag{44}
\end{aligned}
$$

in Theorem C.2. This implies that

$$
\mathbb{E}_{y'|\epsilon} \left[ \left( \frac{p_{\mathcal{M}'(X)}(y'|\epsilon)}{p_{\mathcal{M}'(X')}(y'|\epsilon)} \right)^\alpha \right] \leq \mathbb{E}_{y|\epsilon} \left[ \left( \frac{p_{\mathcal{M}(X)}(y|\epsilon)}{p_{\mathcal{M}(X')}(y|\epsilon)} \right)^\alpha \right].
\tag{45}
$$

Now

$$
\begin{aligned}
(*) &\leq \mathbb{E}_\epsilon \left[ \beta \left( \frac{p_{\mathcal{M}(X)}(\epsilon)}{p_{\mathcal{M}(X')}(\epsilon)} \right)^\alpha \mathbb{E}_{y|\epsilon} \left[ \left( \frac{p_{\mathcal{M}(X)}(y|\epsilon)}{p_{\mathcal{M}(X')}(y|\epsilon)} \right)^\alpha \right] \right] \\
&= \mathbb{E}_{(y,\epsilon) \sim \mathcal{M}(X')} \left[ e^{(1-\alpha)\epsilon} \left( \frac{p_{\mathcal{M}(X)}(y, \epsilon)}{p_{\mathcal{M}(X')}(y, \epsilon)} \right)^\alpha \right] \\
&\leq 1.
\end{aligned}
\tag{46}
$$

$\qquad\square$

**Theorem 4.3.** *Let $\mathcal{M}$ be an $\alpha$-ex-post RDP mechanism with $\mathcal{M}(X)_\epsilon \leq \epsilon_c$ almost surely for all $X \in \mathcal{X}$. Then $\mathcal{M}$ is $(\alpha, \epsilon_c)$-RDP.*

*Proof.* Since $\mathcal{M}(X')_\epsilon \leq \epsilon_c$ almost surely and $\mathcal{M}$ is $\alpha$-unconditionally ex-post RDP, we have for any $X \sim X'$

$$1 \geq \mathop{\mathbb{E}}_{(y,\epsilon)\sim\mathcal{M}(X')} \left[ e^{(1-\alpha)\epsilon} \left( \frac{p_{\mathcal{M}(X)}(y,\epsilon)}{p_{\mathcal{M}(X')}(y,\epsilon)} \right)^\alpha \right] \geq \mathop{\mathbb{E}}_{(y,\epsilon)\sim\mathcal{M}(X')} \left[ e^{(1-\alpha)\epsilon_c} \left( \frac{p_{\mathcal{M}(X)}(y,\epsilon)}{p_{\mathcal{M}(X')}(y,\epsilon)} \right)^\alpha \right]. \tag{47}$$

Since $\mathcal{M}$ returns the pair $(y, \epsilon)$, we must consider their joint distribution in the definition of RDP. Now

$$\frac{1}{\alpha-1} \ln \mathop{\mathbb{E}}_{(y,\epsilon)\sim\mathcal{M}(X')} \left[ \left( \frac{p_{\mathcal{M}(X)}(y,\epsilon)}{p_{\mathcal{M}(X')}(y,\epsilon)} \right)^\alpha \right] \leq \frac{1}{\alpha-1} \ln e^{(\alpha-1)\epsilon_c} = \epsilon_c. \tag{48}$$

$\square$

**Theorem 4.4.** *Let $\mathcal{M}_i(X, y_{1:i-1})$, $1 \leq i \leq T$ be $\alpha$-ex-post RDP mechanisms. Then the adaptive composition $\mathcal{M}^*(X) = (y_{1:T}, \sum_{i=1}^{T} \epsilon_i)$ with*

$$y_i = \mathcal{M}_i(X, y_{1:i-1})_y \tag{12}$$
$$\epsilon_i = \mathcal{M}_i(X, y_{1:i-1})_\epsilon \tag{13}$$

*is $\alpha$-ex-post RDP. If $\mathcal{M}^*$ additionally releases the intermediate $\epsilon_i$ values, it remains $\alpha$-ex-post RDP.*

*Proof.* By post-processing immunity, it suffices to prove the claim for the mechanism releasing the intermediate $\epsilon_i$ values. By induction, proving the claim for $T = 2$ suffices.

Let $\epsilon = \epsilon_1 + \epsilon_2$, $\beta = e^{(1-\alpha)\epsilon}$ and $\beta_i = e^{(1-\alpha)\epsilon_i}$. Starting with the expectation from Definition 4.1, we can remove $\epsilon$ from the expression, since $\epsilon$ is deterministic and does not depend on $X$ when $\epsilon_1, \epsilon_2$ are given. Then we can decompose the expression with the law of total expectation and the chain rule of probabilities:

$$\begin{aligned}
&\mathop{\mathbb{E}}_{(y_1,y_2,\epsilon_1,\epsilon_2,\epsilon)\sim\mathcal{M}^*(X')} \left[ \beta \left( \frac{p_{\mathcal{M}^*(X)}(y_1,y_2,\epsilon_1,\epsilon_2,\epsilon)}{p_{\mathcal{M}^*(X')}(y_1,y_2,\epsilon_1,\epsilon_2,\epsilon)} \right)^\alpha \right] \\
&= \mathop{\mathbb{E}}_{(y_1,y_2,\epsilon_1,\epsilon_2)\sim\mathcal{M}^*(X')} \left[ \beta_1 \beta_2 \left( \frac{p_{\mathcal{M}^*(X)}(y_1,y_2,\epsilon_1,\epsilon_2)}{p_{\mathcal{M}^*(X')}(y_1,y_2,\epsilon_1,\epsilon_2)} \right)^\alpha \right] \\
&= \mathop{\mathbb{E}}_{y_1,\epsilon_1} \left[ \beta_1 \left( \frac{p_{\mathcal{M}_1(X)}(y_1,\epsilon_1)}{p_{\mathcal{M}_1(X')}(y_1,\epsilon_1)} \right)^\alpha \mathop{\mathbb{E}}_{y_2,\epsilon_2|y_1,\epsilon_1} \left[ \beta_2 \left( \frac{p_{\mathcal{M}_2(X)}(y_2,\epsilon_2|y_1,\epsilon_1)}{p_{\mathcal{M}_2(X')}(y_2,\epsilon_2|y_1,\epsilon_1)} \right)^\alpha \right] \right] \\
&= (*)
\end{aligned} \tag{49}$$

By the assumption that $\mathcal{M}_i$ are $\alpha$-unconditionally ex-post RDP,

$$\mathop{\mathbb{E}}_{y_1,\epsilon_1} \left[ \beta_1 \left( \frac{p_{\mathcal{M}_1(X)}(y_1,\epsilon_1)}{p_{\mathcal{M}_1(X')}(y_1,\epsilon_1)} \right)^\alpha \right] \leq 1 \quad \text{and} \tag{50}$$

$$\mathop{\mathbb{E}}_{y_2,\epsilon_2|y_1,\epsilon_1} \left[ \beta_2 \left( \frac{p_{\mathcal{M}_2(X)}(y_2,\epsilon_2|y_1,\epsilon_1)}{p_{\mathcal{M}_2(X')}(y_2,\epsilon_2|y_1,\epsilon_1)} \right)^\alpha \right] \leq 1. \tag{51}$$

This immediately implies that $(*) \leq 1$, which concludes the proof. $\square$

**Theorem 4.5.** *Let $\mathcal{M}_i(X, \epsilon, y_{1:i-1})$, $1 \leq i \leq K$ be mechanisms that are $(\alpha, \epsilon)$-RDP for any $\epsilon \geq 0$. Then the adaptive composition with adaptive privacy bounds $\mathcal{M}^*(X)$ defined as*

$$y_1 = \mathcal{M}_1(X, \epsilon_1, \emptyset) \tag{14}$$
$$\epsilon_i = \mathcal{E}(y_{1:i-1}, \epsilon_{1:i-1}) \tag{15}$$
$$y_i = \mathcal{M}_i(X, \epsilon_i, y_{1:i-1}), \tag{16}$$

*$\mathcal{M}^*(X)_y = y_{1:K}$, and $\mathcal{M}^*(X)_\epsilon = \sum_{i=1}^{K} \epsilon_i$ is $\alpha$-ex-post RDP. The mechanism that additionally releases the intermediate $\epsilon_{1:K}$ values is also $\alpha$-ex-post RDP.*

*Proof.* It suffices the prove the claim for the mechanism releasing the intermediate $\epsilon_{1:K}$ values due to post-processing immunity.

Denote $\beta = e^{(1-\alpha)\epsilon}$ and $\beta_i = e^{(1-\alpha)\epsilon_i}$. Since $\mathcal{M}_i(\cdot, \epsilon_i)$ is $(\alpha, \epsilon_i)$-RDP,

$$\mathbb{E}_{y_i|y_{1:i-1},\epsilon_i} \left[ \left( \frac{p_{\mathcal{M}_i(X)}(y_i|y_{1:i-1}, \epsilon_i)}{p_{\mathcal{M}_i(X')}(y_i|y_{1:i-1}\epsilon_i)} \right)^\alpha \right] \le e^{\epsilon_i(\alpha-1)} = \beta_i^{-1}. \tag{52}$$

The functions computing $\epsilon_i$ and $\epsilon$ do not depend on $X$, so we can cancel them from the likelihood ratio:

$$\mathbb{E}_{y_{1:K},\epsilon_{1:K},\epsilon} \left[ \beta \left( \frac{p_{\mathcal{M}^*(X)}(y_{1:K}, \epsilon_{1:K}, \epsilon)}{p_{\mathcal{M}^*(X')}(y_{1:K}, \epsilon_{1:K}, \epsilon)} \right)^\alpha \right]$$

$$= \mathbb{E}_{y_{1:K},\epsilon_{1:K},\epsilon} \left[ \beta \left( \frac{p_\epsilon(\epsilon|\epsilon_{1:K})}{p_\epsilon(\epsilon|\epsilon_{1:K})} \prod_{i=1}^K \frac{p_{\mathcal{M}_i(X)}(y_i|y_{1:i-1}, \epsilon_{1:i-1}) p_\mathcal{E}(\epsilon_i|y_{1:i-1}, \epsilon_{1:i-1})}{p_{\mathcal{M}_i(X')}(y_i|y_{1:i-1}, \epsilon_{1:i-1}) p_\mathcal{E}(\epsilon_i|y_{1:i-1}, \epsilon_{1:i-1})} \right)^\alpha \right] \tag{53}$$

$$= \mathbb{E}_{y_{1:K},\epsilon_{1:K},\epsilon} \left[ \beta \left( \prod_{i=1}^K \frac{p_{\mathcal{M}_i(X)}(y_i|y_{1:i-1}, \epsilon_i)}{p_{\mathcal{M}_i(X')}(y_i|y_{1:i-1}, \epsilon_i)} \right)^\alpha \right]$$

$$= (*).$$

Since $\epsilon$ is deterministic given $\epsilon_{1:K}$, we can remove the expectation over $\epsilon$:

$$(*) = \mathbb{E}_{y_{1:K},\epsilon_{1:K}} \left[ e^{(1-\alpha)\sum_{i=1}^K \epsilon_i} \left( \prod_{i=1}^K \frac{p_{\mathcal{M}_i(X)}(y_i|y_{1:i-1}, \epsilon_i)}{p_{\mathcal{M}_i(X')}(y_i|y_{1:i-1}, \epsilon_i)} \right)^\alpha \right]$$

$$= \mathbb{E}_{y_{1:K},\epsilon_{1:K}} \left[ \prod_{i=1}^K \beta_i \left( \frac{p_{\mathcal{M}_i(X)}(y_i|y_{1:i-1}, \epsilon_i)}{p_{\mathcal{M}_i(X)}(y_i|y_{1:i-1}, \epsilon_i)} \right)^\alpha \right]. \tag{54}$$

Next, we use the law of total expectation and pull out known quantities from the resulting inner expectation:

$$(*) = \mathbb{E}_{y_{1:K-1},\epsilon_{1:K-1}} \left[ \prod_{i=1}^{K-1} \beta_i \left( \frac{p_{\mathcal{M}_i(X)}(y_i|y_{1:i-1}, \epsilon_i)}{p_{\mathcal{M}_i(X)}(y_i|y_{1:i-1}\epsilon_i)} \right)^\alpha \right.$$

$$\left. \mathbb{E}_{y_K,\epsilon_K|y_{1:K-1},\epsilon_{1:K-1}} \left[ \beta_K \left( \frac{p_{\mathcal{M}_K(X)}(y_K|y_{1:K-1}, \epsilon_K)}{p_{\mathcal{M}_K(X')}(y_K|y_{1:K-1}, \epsilon_K)} \right)^\alpha \right] \right]. \tag{55}$$

Looking at the inner expectation, we use the law of total expectation and the $(\alpha, \epsilon_K)$-RDP of $\mathcal{M}_K$:

$$\mathbb{E}_{y_K,\epsilon_K|y_{1:K-1},\epsilon_{1:K-1}} \left[ \beta_K \left( \frac{p_{\mathcal{M}_K(X)}(y_K|y_{1:K-1}, \epsilon_K)}{p_{\mathcal{M}_K(X')}(y_K|y_{1:K-1}, \epsilon_K)} \right)^\alpha \right]$$

$$= \mathbb{E}_{\epsilon_K|y_{1:K-1},\epsilon_{1:K-1}} \left[ \beta_K \mathbb{E}_{y_K|y_{1:K-1},\epsilon_K} \left[ \left( \frac{p_{\mathcal{M}_K(X)}(y_K|y_{1:K-1}, \epsilon_K)}{p_{\mathcal{M}_K(X')}(y_K|y_{1:K-1}, \epsilon_K)} \right)^\alpha \right] \right] \tag{56}$$

$$\le \mathbb{E}_{\epsilon_K|y_{1:K-1},\epsilon_{1:K-1}} \left[ \beta_K \beta_K^{-1} \right]$$

$$= 1.$$

Plugging into $(*)$ gives

$$(*) \le \mathbb{E}_{y_{1:K-1},\epsilon_{1:K-1}} \left[ \prod_{i=1}^{K-1} \beta_i \left( \frac{p_{\mathcal{M}_i(X)}(y_i|\epsilon_i)}{p_{\mathcal{M}_i(X)}(y_i|\epsilon_i)} \right)^\alpha \right]. \tag{57}$$

Repeating these steps an additional $K - 1$ times results in

$$(*) \le 1, \tag{58}$$

which concludes the proof. $\qquad\square$

---

**Algorithm 2:** Ex-post Composition with Data-Dependent Stopping Criterion with Output Padded to Length $K$

---

**Input:** Dataset $X$ divided into train set $X_y$ and validation set $X_\epsilon$, $(\alpha, \epsilon_i)$-RDP mechanisms $\mathcal{M}_i$, $(\alpha, \epsilon_\mathcal{T})$-RDP stopping
      rule $\mathcal{T}$, privacy bound choosing rule $\mathcal{E}$

**Output:** Sequence of outputs $y_{1:K}$, privacy bounds $\epsilon_{1:K}$, total privacy bound $\epsilon$

has_halted $\leftarrow$ False;

**for** $1 \le i \le K$ **do**
    | **if** *not* has_halted **then**
    |     | $\epsilon_i \leftarrow \mathcal{E}(y_{1:i-1}, \epsilon_{i:1-1})$;
    |     | $y_i \leftarrow \mathcal{M}_i(X_y, \epsilon_i, y_{1:i-1})$;
    |     | $c \leftarrow \mathcal{T}(X_\epsilon, y_i)$;
    |     | **if** $c = $ Halt **then**
    |     |     | has_halted $\leftarrow$ True;
    |     | **end**
    | **else**
    |     | $\epsilon_i \leftarrow 0$;
    |     | $y_i \leftarrow \perp$;
**end**

$\epsilon \leftarrow \max\left(\sum_{j=1}^K \epsilon_j, \epsilon_\mathcal{T}\right)$;

**return** $y_{1:K}, \epsilon_{1:K}, \epsilon$;

---

**Theorem 4.6.** *Let a dataset $X$ be divided into a training set $X_y$ and validation set $X_\epsilon$, that are disjoint, in any data-independent manner that ensures any neighbouring datasets $X_1 \sim X_2$ only differ in one part of the split. Let $\mathcal{M}_i(X_y, \epsilon_i, y_{1:i-1})$, $1 \le i \le K$ be mechanisms that are $(\alpha, \epsilon_i)$-RDP, and let $\mathcal{T}(X_\epsilon, y_{1:i-1})$ be a stopping rule that is $(\alpha, \epsilon_\mathcal{T})$-RDP, $\alpha > 1$. Then the adaptive composition with adaptive data-dependent privacy bounds $\mathcal{M}^*$ defined in Algorithm 1 with $\mathcal{M}^*(X)_y = (t, y_{1:t}, \epsilon_{1:t})$, $\mathcal{M}^*(X)_\epsilon = \epsilon$ is $\alpha$-ex-post RDP.*

*Proof.* We consider an equivalent mechanism that pads the sequences $y_{1:t}$ and $\epsilon_{1:t}$ to length $K$ with $y_i = \perp$ and $\epsilon_i = 0$ for $i > t$, where $\perp$ is a marker for missing values, shown in Algorithm 2. Note that this makes $\epsilon_i$ given $y_{1:i-1}, \epsilon_{1:i-1}$ depend on the private data $X_\epsilon$, as $\epsilon_i$ depends on whether the mechanism halted on an iteration before $i$. We denote $\beta = e^{(1-\alpha)\epsilon}$. Using the chain rule of probability and cancelling the probability ratio of releasing $\epsilon$ which does not depend on $X$, we have

$$
\mathbb{E}_{y_{1:K}, \epsilon_{1:K}, \epsilon}\left[\beta\left(\frac{p_{\mathcal{M}^*(X)}(y_{1:K}, \epsilon_{1:K}, \epsilon)}{p_{\mathcal{M}^*(X')}(y_{1:K}, \epsilon_{1:K}, \epsilon)}\right)^\alpha\right]
$$

$$
= \mathbb{E}_{y_{1:K}, \epsilon_{1:K}, \epsilon}\left[\beta\left(\frac{p_\epsilon(\epsilon|\epsilon_{1:t})}{p_\epsilon(\epsilon|\epsilon_{1:t})}\prod_{i=1}^K \frac{p_{\mathcal{E}(X_\epsilon)}(\epsilon_i|y_{1:i-1}, \epsilon_{1:i-1})p_{\mathcal{M}_i(X_y)}(y_i|\epsilon_i, y_{1:i-1})}{p_{\mathcal{E}(X'_\epsilon)}(\epsilon_i|y_{1:i-1}, \epsilon_{1:i-1})p_{\mathcal{M}_i(X'_y)}(y_i|\epsilon_i, y_{1:i-1})}\right)^\alpha\right] \qquad (59)
$$

$$
= \mathbb{E}_{y_{1:K}, \epsilon_{1:K}, \epsilon}\left[\beta\left(\prod_{i=1}^K \frac{p_{\mathcal{E}(X_\epsilon)}(\epsilon_i|y_{1:i-1}, \epsilon_{1:i-1})p_{\mathcal{M}_i(X_y)}(y_i|\epsilon_i, y_{1:i-1})}{p_{\mathcal{E}(X'_\epsilon)}(\epsilon_i|y_{1:i-1}, \epsilon_{1:i-1})p_{\mathcal{M}_i(X'_y)}(y_i|\epsilon_i, y_{1:i-1})}\right)^\alpha\right]
$$

$$
= (*).
$$

Since $X_y$ and $X_\epsilon$ are disjoint, we have that $X_y = X'_y$ or $X_\epsilon = X'_\epsilon$. In both cases, we can cancel the related probability

ratios. We can also remove $\epsilon$ from the expectation, since it is deterministic given $\epsilon_{1:K}$. We start with the case $X_y = X'_y$:

$$
\begin{aligned}
(*) &= \mathop{\mathbb{E}}_{y_{1:K}, \epsilon_{1:K}, \epsilon} \left[ \beta \left( \prod_{i=1}^{K} \frac{p_{\mathcal{E}(X_\epsilon)}(\epsilon_i | y_{1:i-1}, \epsilon_{1:i-1})}{p_{\mathcal{E}(X'_\epsilon)}(\epsilon_i | y_{1:i-1}, \epsilon_{1:i-1})} \right)^{\alpha} \right] \\
&= \mathop{\mathbb{E}}_{y_{1:K}, \epsilon_{1:K}} \left[ e^{(1-\alpha) \max(\sum_{i=1}^{K} \epsilon_i, \epsilon_\mathcal{T})} \left( \prod_{i=1}^{K} \frac{p_{\mathcal{E}(X_\epsilon)}(\epsilon_i | y_{1:i-1}, \epsilon_{1:i-1})}{p_{\mathcal{E}(X'_\epsilon)}(\epsilon_i | y_{1:i-1}, \epsilon_{1:i-1})} \right)^{\alpha} \right] \\
&\leq \mathop{\mathbb{E}}_{y_{1:K}, \epsilon_{1:K}} \left[ e^{(1-\alpha)\epsilon_\mathcal{T}} \left( \prod_{i=1}^{K} \frac{p_{\mathcal{E}(X_\epsilon)}(\epsilon_i | y_{1:i-1}, \epsilon_{1:i-1})}{p_{\mathcal{E}(X'_\epsilon)}(\epsilon_i | y_{1:i-1}, \epsilon_{1:i-1})} \right)^{\alpha} \right] \\
&\leq 1.
\end{aligned}
\tag{60}
$$

The final inequality uses the assumption that $\mathcal{T}$ is $(\alpha, \epsilon_\mathcal{T})$-RDP.

In the case that $X_\epsilon = X'_\epsilon$:

$$
(*) = \mathop{\mathbb{E}}_{y_{1:K}, \epsilon_{1:K}, \epsilon} \left[ \beta \left( \prod_{i=1}^{K} \frac{p_{\mathcal{M}_i(X_y)}(y_i | \epsilon_i, y_{1:i-1})}{p_{\mathcal{M}_i(X'_y)}(y_i | \epsilon_i, y_{1:i-1})} \right)^{\alpha} \right]
\tag{61}
$$

Now we can follow the steps in the proof of Theorem 4.5 after Equation (53) to show that $(*) \leq 1$.

$\square$

## C.4. Ex-post RDP of the Brownian Mechanism

**Definition C.3** (Bass, 2011). *A stochastic process $B_t$ is a Brownian motion started at $y$, with respect to a filtration $\{\mathcal{F}_t\}$, if*

1. *$B_t$ is $\mathcal{F}_t$-measurable*

2. *$B_0 = y$*

3. *$B_t - B_s \sim \mathcal{N}(0, t - s)$ for $s < t$*

4. *$B_t - B_s$ is independent of $\mathcal{F}_s$ when $s < t$*

5. *$B_t$ has continuous paths*

A *standard* Brownian motion has $B_0 = 0$. A $d$-dimensional Brownian motion is a vector of $d$ independent one-dimensional Brownian motions.

**Lemma 5.1.** *The sequential precision-weighted Gaussian mechanism (Algorithm 3) is $\alpha$-ex-post RDP.*

*Proof.* Algorithm 3 is almost a post-processing of a composition of Gaussian mechanisms, suitable for applying Theorem 4.5. The only difference is that in Algorithm 3, $\mathcal{E}$ takes $\hat{s}_{1:i-1}$ as input when computing $\epsilon_i$, which does not fit the assumptions of Theorem 4.5. However, we can define an equivalent $\epsilon$ selector $\mathcal{E}'$ that takes $\tilde{s}_{1:i-1}$ and $\epsilon_{1:i-1}$ as input, computes $\hat{s}_{1:i-1}$, and returns $\mathcal{E}(\hat{s}_{1:i-1}, \epsilon_{1:i-1})$, and adjust Algorithm 3 to use $\mathcal{E}'$ instead of $\mathcal{E}$. This adjustment does not change the outputs of the algorithm, so any privacy bound we prove applies to Algorithm 3, but it allows us to apply Theorem 4.5.

The Gaussian mechanisms have RDP bounds $(\alpha, \epsilon_i - \epsilon_{i-1})$ for the $i$th mechanism. The sum of these $\epsilon$ values is $\epsilon_1 + \sum_{i=2}^{K} (\epsilon_i - \epsilon_{i-1}) = \epsilon_K$, so the claim follows from Theorems 4.5 and 4.2. $\square$

**Theorem 5.2.** *For any dataset $X$, function $f$ with $\ell_2$-sensitivity $\Delta$, RDP order $\alpha$, $\epsilon$ selector $\mathcal{E}$, the output of Algorithm 3 has the same distribution as the Brownian mechanism in Algorithm 4.*

*Proof.* Denote the $\hat{s}_{1:K}$ outputs from Algorithm 3 as $\hat{s}_{1:K}^{\text{seq}}$ and the outputs from Algorithm 4 as $\hat{s}_{1:K}^{\text{bro}}$.

---

**Algorithm 3:** Sequential Precision-Weighted Gaussian Mechanism

---

**Input:** Data $X$, function $f$, $\ell_2$-sensitivity $\Delta$, RDP order $\alpha$, $\epsilon$ selector $\mathcal{E}$.

**Output:** Noisy estimates $\hat{s}_{1:K}$, privacy bounds $\epsilon_{1:K}$

$\epsilon_1 \leftarrow \mathcal{E}(\emptyset, \emptyset)$;

$\sigma_1^2 \leftarrow \frac{\alpha \Delta^2}{2\epsilon_1}$;

$\tilde{s}_1 \leftarrow f(X) + \mathcal{N}(0, \sigma_1^2)$;

$\hat{s}_1 \leftarrow \tilde{s}_1$;

**for** $2 \leq i \leq K$ **do**

$\qquad \epsilon_i \leftarrow \mathcal{E}(\hat{s}_{1:i-1}, \epsilon_{1:i-1})$;

$\qquad \sigma_i^2 \leftarrow \frac{\alpha \Delta^2}{2(\epsilon_i - \epsilon_{i-1})}$;

$\qquad \tilde{s}_i \leftarrow f(X) + \mathcal{N}(0, \sigma_i^2)$;

$\qquad \hat{s}_i \leftarrow \frac{1}{\sum_{j=1}^{i} \frac{1}{\sigma_j^2}} \sum_{j=1}^{i} \frac{1}{\sigma_j^2} \tilde{s}_j$;

**end**

**return** $\hat{s}_{1:K}, \epsilon_{1:K}$;

---

**Algorithm 4:** Brownian Mechanism

---

**Input:** Data $X$, function $f$, $\ell_2$-sensitivity $\Delta$, RDP order $\alpha$, $\epsilon$ selector $\mathcal{E}$.

**Output:** Noisy estimates $\hat{s}_{1:K}$, privacy bounds $\epsilon_{1:K}$

$\epsilon_1 \leftarrow \mathcal{E}(\emptyset, \emptyset)$;

$T_1 \leftarrow \frac{\alpha \Delta^2}{2\epsilon_1}$;

Sample $B_{T_1} \sim \mathcal{N}(0, T_1)$;

$\hat{s}_1 \leftarrow f(X) + B_{T_1}$;

**for** $2 \leq i \leq K$ **do**

$\qquad \epsilon_i \leftarrow \mathcal{E}(\hat{s}_{1:i-1}, \epsilon_{1:i-1})$;

$\qquad T_i \leftarrow \frac{\alpha \Delta^2}{2\epsilon_i}$;

$\qquad$ Sample $B_{T_i}$ from standard Brownian motion conditional on $B_{T_1}, \ldots, B_{T_{i-1}}$;

$\qquad \hat{s}_i \leftarrow f(X) + B_{T_i}$;

**end**

**return** $\hat{s}_{1:K}, \epsilon_{1:K}$;

---

In the Brownian mechanism (Whitehouse et al., 2022), $\hat{s}_1^{\mathrm{bro}}|X \sim \mathcal{N}(f(X), T_1)$. When $T_i < T_{i-1}$,

$$\hat{s}_i^{\mathrm{bro}}|\hat{s}_1^{\mathrm{bro}}, \ldots, \hat{s}_{i-1}^{\mathrm{bro}}, X \sim \mathcal{N}\left(f(X) + \frac{T_i}{T_{i-1}}(\hat{s}_{i-1}^{\mathrm{bro}} - f(X)), \frac{(T_{i-1} - T_i)T_i}{T_{i-1}}\right). \tag{62}$$

We show that the outputs of Algorithm 3 have these same conditional distributions. Clearly $\hat{s}_1^{\mathrm{seq}}|X \sim \mathcal{N}(f(X), \sigma_1^2)$. Since $\sigma_1^2 = \frac{\alpha \Delta^2}{2\epsilon_1} = T_1$, $\hat{s}_1^{\mathrm{seq}}$ has the same distribution as $\hat{s}_1^{\mathrm{bro}}$.

For $2 \leq i \leq K$,

$$\left(\sum_{j=1}^i \frac{1}{\sigma_i^2}\right)^{-1} = \left(\sum_{j=2}^i \frac{2(\epsilon_j - \epsilon_{j-1})}{\alpha \Delta^2} + \frac{2\epsilon_1}{\alpha \Delta^2}\right)^{-1} = \left(\frac{2\epsilon_i^2}{\alpha \Delta^2}\right)^{-1} = T_i \tag{63}$$

and

$$\hat{s}_i^{\mathrm{seq}} = T_i \sum_{j=1}^i \frac{1}{\sigma_j^2}\tilde{s}_j = T_i \left(\sum_{j=1}^{i-1} \frac{1}{\sigma_j^2}\tilde{s}_j + \frac{1}{\sigma_i^2}\tilde{s}_i\right) = T_i \left(\frac{1}{T_{i-1}}\hat{s}_{i-1}^{\mathrm{seq}} + \frac{1}{\sigma_i^2}\tilde{s}_i\right). \tag{64}$$

Since $\hat{s}_i^{\mathrm{seq}}$ is a linear function of $\tilde{s}_i$, the distribution $\hat{s}_i^{\mathrm{seq}}|\hat{s}_1^{\mathrm{seq}}, \ldots, \hat{s}_{i-1}^{\mathrm{seq}}, X$ is Gaussian. The mean is

$$\begin{aligned}
\mathbb{E}(\hat{s}_i^{\mathrm{seq}}|\hat{s}_1^{\mathrm{seq}}, \ldots, \hat{s}_{i-1}^{\mathrm{seq}}, X) &= T_i \left(\frac{1}{T_{i-1}}\hat{s}_{i-1}^{\mathrm{seq}} + \frac{1}{\sigma_i^2}\mathbb{E}(\tilde{s}_i|X)\right) \\
&= T_i \left(\frac{1}{T_{i-1}}\hat{s}_{i-1}^{\mathrm{seq}} + \frac{1}{\sigma_i^2}f(X)\right) \\
&= \frac{T_i}{T_{i-1}}\hat{s}_{i-1}^{\mathrm{seq}} + \frac{T_i}{\sigma_i^2}f(X) \\
&= \frac{T_i}{T_{i-1}}\hat{s}_{i-1}^{\mathrm{seq}} + \frac{\epsilon_i - \epsilon_{i-1}}{\epsilon_i}f(X) \\
&= f(X) + \frac{T_i}{T_{i-1}}\hat{s}_{i-1}^{\mathrm{seq}} - \frac{\epsilon_{i-1}}{\epsilon_i}f(X) \\
&= f(X) + \frac{T_i}{T_{i-1}}(\hat{s}_{i-1}^{\mathrm{seq}} - f(X))
\end{aligned} \tag{65}$$

and the variance is

$$\begin{aligned}
\mathrm{Var}(\hat{s}_i^{\mathrm{seq}}|\hat{s}_1^{\mathrm{seq}}, \ldots, \hat{s}_{i-1}^{\mathrm{seq}}) &= T_i^2 \cdot \frac{1}{\sigma_i^4}\mathrm{Var}(\tilde{s}_i) = T_i^2 \cdot \frac{1}{\sigma_i^2} = T_i \cdot \frac{T_i}{\sigma_i^2} \\
&= T_i \cdot \frac{\epsilon_i - \epsilon_{i-1}}{\epsilon_i} = T_i \cdot \left(1 - \frac{\epsilon_{i-1}}{\epsilon_i}\right) \\
&= T_i \cdot \frac{T_{i-1} - T_i}{T_{i-1}}.
\end{aligned} \tag{66}$$

These match the Brownian mechanism. This means that the output distributions of the Brownian mechanism and Algorithm 3 are identical. $\square$

## D. Experiment Details and Extra Results

**Synthetic Data Details** The summaries of real data we use are *marginal queries*, which count how many rows of a discrete dataset have given values for given variables. While marginal queries are very simple, the Private-PGM algorithm (McKenna et al., 2019) is able to generate high quality synthetic data based on only noisy values of marginal queries (Tao et al., 2021; Chen et al., 2025). To choose the marginal queries, we use the MST algorithm (McKenna et al., 2021), to obtain an initial set, and add the pairwise marginal queries between the classification target and all other variables. The sensitivity of a

single marginal query under add/remove neighbourhood is 1 (McKenna et al., 2021), so the $\ell_2$-sensitivity for $n_q$ queries is $\Delta = \sqrt{n_q}$.

As we randomly split the data into train and validation sets, we use Theorem 4.6 to prove that our synthetic data generator is ex-post RDP. Since we already know the Brownian mechanism is ex-post RDP, we only need to ensure the accuracy check is RDP. Both check mechanisms depend on the sensitivity of the accuracy, which is $\Delta_{\mathrm{acc}} = 1/n_{\mathrm{validation}}$ under add/remove neighbourhood.

**Plain Gaussian Details**    The plain Gaussian mechanism simply releases the accuracies with Gaussian noise. To account for multiple checks we divide the privacy budget with the maximum number of checks, which is $m-1$ if there are $m$ possible $\epsilon$ values. This ensures that the composition of all checks is $\epsilon_{\mathrm{check}}$-RDP. This means the noise variance is $\frac{\alpha \Delta_{\mathrm{acc}}^2 (m-1)}{2\epsilon_{\mathrm{check}}}$.

**SVT Details**    For SVT, we use Theorem 8, eq. (3) from Zhu & Wang (2020), which requires us to add Gaussian noise to the threshold, and Laplace noise to the accuracy values. We also examined two other results from Zhu & Wang (2020, Remark (Bounded-length SVT), Proposition 10) that would be applicable, but found that they have too large overheads in privacy cost to be useful in our setting.

Denote the variance of the Gaussian by $\sigma_1^2$, and the variance of the Laplace by $\sigma_2^2$. Let $t \in (0,1)$ be a variable controlling the variance split such that the Gaussian mechanism has a $t$ fraction of the total variance, meaning

$$\sigma_1^2 = t(\sigma_1^2 + \sigma_2^2). \tag{67}$$

To make the Gaussian $\epsilon_1$-RDP for sensitivity $\Delta_{\mathrm{acc}}$ and the Laplace $\epsilon_2$-DP for sensitivity $2\Delta_{\mathrm{acc}}$, as required by the theorem, we need

$$\epsilon_1 = \frac{\alpha \Delta_{\mathrm{acc}}^2}{2\sigma_1^2} \qquad \epsilon_2 = \frac{2\sqrt{2}\Delta_{\mathrm{acc}}}{\sigma_2}. \tag{68}$$

The theorem requires $\epsilon_1 + \epsilon_2 = \epsilon_{\mathrm{check}}$. Plugging in (68), the solution of $\sigma_2$ from (67) and rearranging, we obtain the quadratic equation

$$\epsilon_{\mathrm{check}}\sigma_1^2 - \frac{2\sqrt{2}\Delta_{\mathrm{acc}}}{t'}\sigma_1 - \frac{\alpha \Delta_{\mathrm{acc}}^2}{2} = 0 \tag{69}$$

where $t' = \sqrt{\frac{1-t}{t}}$. The equation has a unique positive solution, which we set $\sigma_1$ to. We then solve $\sigma_2$ from (67). Finally, we numerically optimise $t$ to minimise the total variance $\sigma_1^2 + \sigma_2^2$.

SVT permits an arbitrary number of accuracy checks, unlike the plain Gaussian check. However, SVT has a large overhead in the noise variance with a small number of checks, so in our setting, the plain Gaussian check requires less noise.

**Dataset and Preprocessing**    We use the UCI Adult dataset (Kohavi & Becker, 1996). We remove the columns "fnlwgt" and "educational-num", as the first one is a weighting and the second contains the same information as another column. We remove rows with missing values. We discretise the "age" column to 8 categories, with edges $10, 20, \ldots, 90$, and "hours-per-week" to 10 categories with edges $0, 10, 20, \ldots, 90$, including a category for $\geq 90$. We discretise "capital-gain" and "capital-loss" to binary values indicating whether the value is positive. After these preprocessing steps, the dataset has 45222 rows and 13 columns.

We use a 40%-40%-20% train-validation-test split, with the validation-test split stratified such that the test and validation sets have the same proportions of both labels. The split is fixed for all repeats of the experiment. The validation accuracy of the gradient boosting classifier on synthetic data without DP is $0.829 \pm 0.0014$, while the test accuracy is $0.829 \pm 0.0011$ (average $\pm$ standard deviation over 10 runs). Based on preliminary runs, we set an accuracy threshold of 0.825, which the mechanism is capable of reaching with high $\epsilon$, but not low $\epsilon$.

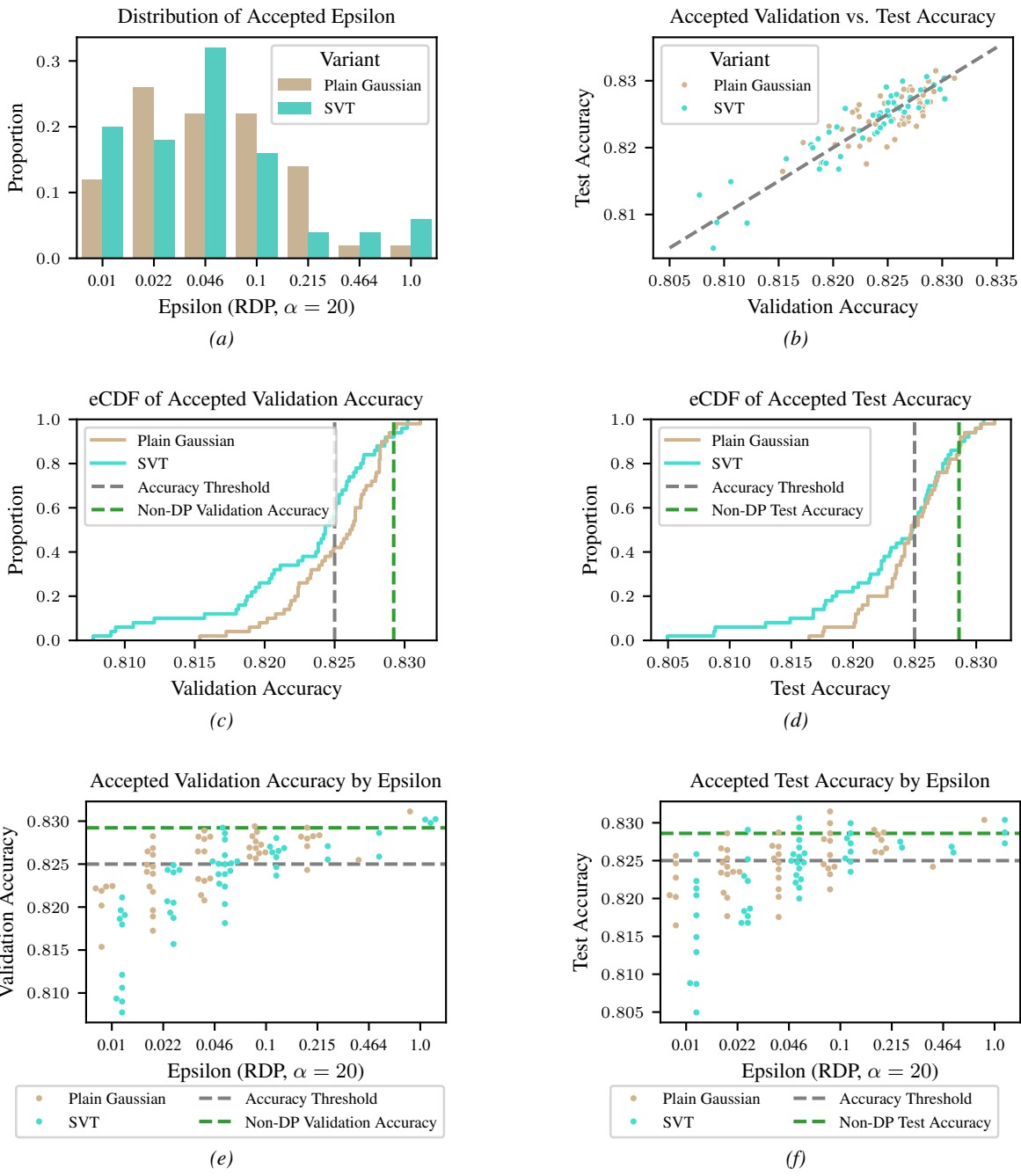

*Figure 3.* (a) Distribution of accepted $\epsilon$ values. SVT generally accepts earlier, resulting in lower $\epsilon$. (b) Scatterplot of validation and test accuracies of accepted results. There is no indication of overfitting to the validation set, as the points cluster around the diagonal gray line. (c, d) Empirical CDFs of accepted validation (c) and test (d) accuracies. Plain Gaussian gives higher accuracies. The validation accuracies from Plain Gaussian are higher than the threshold (60%) more often than the validation accuracies from SVT (40%). (e, f) Validation (e) and test (f) accuracies by $\epsilon$. Plain Gaussian has generally higher accuracies for each $\epsilon$. In all plots, non-DP validation and test accuracies are computed from synthetic data without DP noise. The plotted value is a mean over 10 repeats of synthetic data generation.

*Table 3.* Conversion of ex-post RDP $\epsilon$ values used for synthetic data generation to approximate ex-post privacy $\epsilon$ with $\delta = 10^{-5}$ using the formula of Ghazi et al. (2025, Lemma 7)

| RDP Epsilon @ $\alpha = 20$ | ADP Epsilon @ $\delta = 10^{-5}$ |
|---|---|
| 0.010000 | 0.615943 |
| 0.021544 | 0.627488 |
| 0.046416 | 0.652359 |
| 0.100000 | 0.705943 |
| 0.215443 | 0.821387 |
| 0.464159 | 1.070102 |
| 1.000000 | 1.605943 |

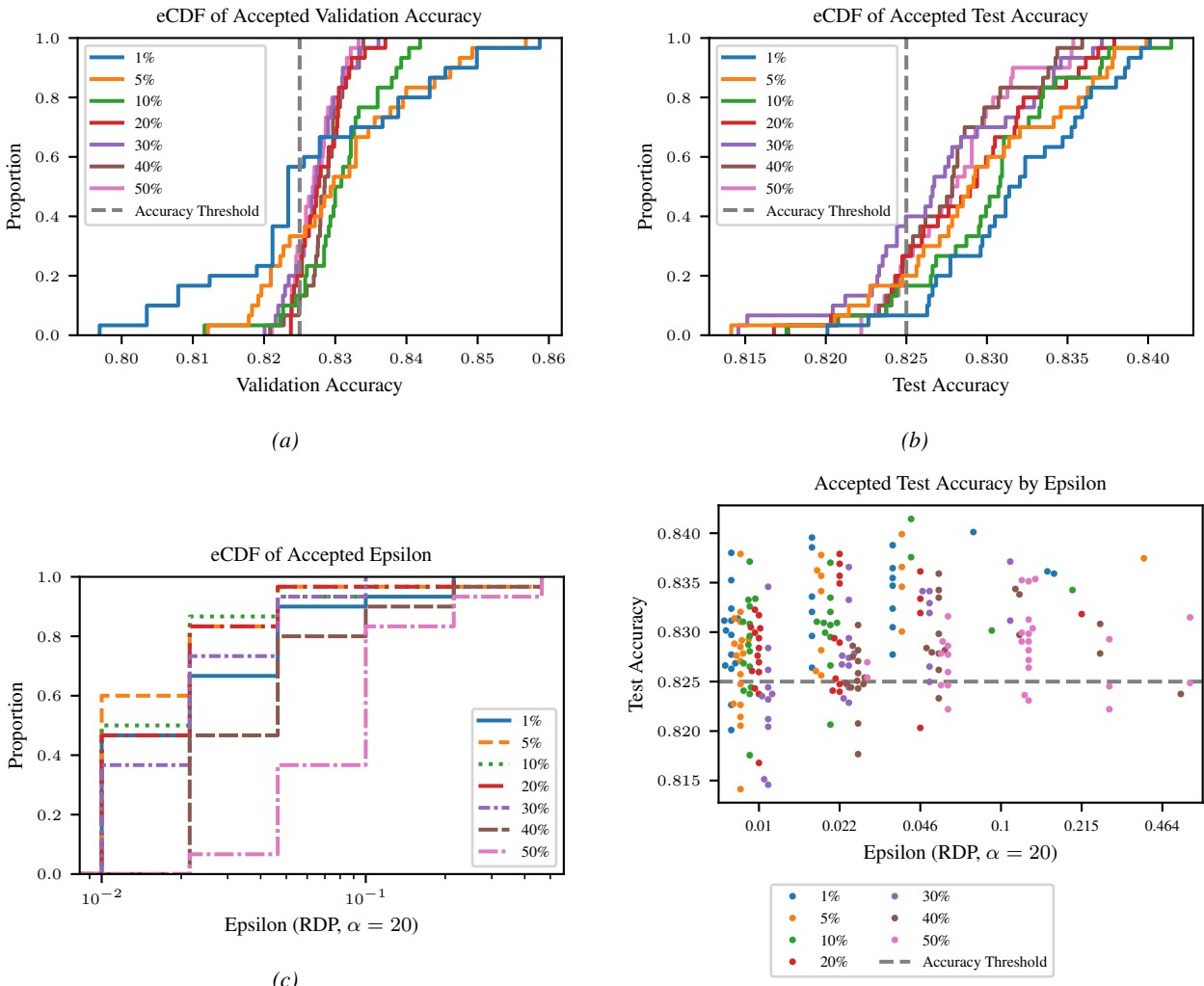

*Figure 4.* A repeat of the synthetic data generation experiment in Section 6, varying the size of the validation dataset. Each line corresponds to a different validation set size, given in percentage of the whole dataset. The test dataset is 20% of the whole. The results show that a smaller validation set improves test accuracy, but at the cost of a higher final $\epsilon$ for the very small sizes.

## D.1. Varying Validation Dataset Size

In this section, we investigate how the size of the validation set affects the results in the synthetic data generation experiment from Section 6. We compare validation data proportions between 1% and 50% (in Section 6 we used 40%). The test set is 20% of the full data, and the rest is the training set. Unlike the main experiment, we create the splits independently in each repeated run. We reduce the number of repeated runs to 30, but otherwise keep the same setup.

The results in Figure 4 and Table 4 show that a smaller validation set improves test accuracy, but at the cost of a higher final $\epsilon$ for the very small sizes. The smallest 1% size also increases the probability of accepting with a validation accuracy below the threshold, though without a corresponding decrease in test accuracy. This is likely caused by the increased variance with the very small validation set – the validation accuracy is small because the validation set happened to have more hard cases, but the synthetic data generated from the larger training set was good, giving good test accuracy.

*Table 4.* Proportion of repeats where the first validation accuracy crossing the threshold was accepted for the new experiment in Figure 4
The proportion is lower for the two smallest validation proportions. The standard deviation is divided by the $\sqrt{n_{repeats}} = \sqrt{30}$.

| Validation Percentage | mean | std |
|---|---|---|
| 1% | 0.233 | 0.079 |
| 5% | 0.433 | 0.092 |
| 10% | 0.533 | 0.093 |
| 20% | 0.567 | 0.092 |
| 30% | 0.633 | 0.089 |
| 40% | 0.700 | 0.085 |
| 50% | 0.467 | 0.093 |

## D.2. Alternative Privacy Bounds

In this section, we detail how the privacy bounds would change if we used $\alpha'$ instead of $\alpha = 20$. For the Brownian mechanism:

$$\epsilon(\alpha') = \epsilon/\alpha \cdot \alpha' \tag{70}$$

where $\epsilon$ is the privacy bound from Figure 2. For the plain Gaussian accuracy check + the Brownian mechanism:

$$\epsilon_{\text{total}}(\alpha') = (\epsilon + \epsilon_{\text{query}})/\alpha \cdot \alpha' \tag{71}$$

where $\epsilon_{\text{query}} = 0.01$.

For the SVT + Brownian mechanism:

$$
\begin{aligned}
\epsilon_{\text{total}}(\alpha') &= \frac{2\sqrt{2}\Delta_{\text{acc}}}{\sigma_2} + \frac{\Delta_{\text{acc}}^2}{2\sigma_1^2}\alpha' + \epsilon/\alpha \cdot \alpha' \\
&\approx 9.292 \cdot 10^{-3} + (3.541 \cdot 10^{-5} + \epsilon/\alpha) \cdot \alpha'
\end{aligned}
\tag{72}
$$

where $\Delta_{\text{acc}} = 1/n_{\text{validation}}$ is the sensitivity of the accuracy, $\sigma_1$ and $\sigma_2$ are the noise scales inside the SVT.

We can express these bounds with an ex-post variant of zero-concentrated DP (zCDP; Bun & Steinke 2016). First, we present the ex-ante zCDP definition.

**Definition D.1** (Bun & Steinke, 2016, Definition 1). *A mechanism $\mathcal{M}\colon \mathcal{X} \to \mathcal{Y}$ is $(\xi, \rho)$-zCDP if, for all neighbouring $X \sim X'$ and all $\alpha > 1$,*

$$\frac{1}{\alpha - 1} \ln \underset{y \sim \mathcal{M}(X')}{\mathbb{E}} \left[ \left( \frac{p_{\mathcal{M}(X)}(y)}{p_{\mathcal{M}(X')}(y)} \right)^\alpha \right] \leq \xi + \rho\alpha. \tag{73}$$

Next, we present ex-post zCDP. We take $\rho$ to be the privacy parameter that is returned by the mechanism, and make $\xi$ fixed.

**Definition D.2.** *A mechanism $\mathcal{M}\colon \mathcal{X} \to \mathcal{Y} \times \mathbb{R}_{\geq 0}$ is $\xi$-ex-post zCDP if, for all neighbouring $X \sim X'$ and all $\alpha > 1$,*

$$\underset{(y,\rho) \sim \mathcal{M}(X')}{\mathbb{E}} \left[ e^{(1-\alpha)(\xi+\rho\alpha)} \left( \frac{p_{\mathcal{M}(X)}(y,\rho)}{p_{\mathcal{M}(X')}(y,\rho)} \right)^\alpha \right] \leq 1. \tag{74}$$

We will use $\mathcal{M}(X)_\rho$ to denote the returned $\rho$-value from an ex-post zCDP mechanism $\mathcal{M}$ instead of $\mathcal{M}(X)_\epsilon$.

The following lemma characterises ex-post zCDP in terms of ex-post RDP.

**Lemma D.3.** *A mechanism $\mathcal{M}\colon \mathcal{X} \to \mathcal{Y} \times \mathbb{R}_{\geq 0}$ is $\xi$-ex-post zCDP if and only if the mechanism $\mathcal{M}_{\text{RDP}}$ defined as $\mathcal{M}_{\text{RDP}}(X)_y = \mathcal{M}(X)_y$ and $\mathcal{M}_{\text{RDP}}(X)_\epsilon = \xi + \alpha\mathcal{M}(X)_\rho$ is $\alpha$-ex-post RDP for all $\alpha > 1$.*

*Proof.* Due to the definition of $\mathcal{M}_{\text{RDP}}$, we have

$$\underset{(y,\rho) \sim \mathcal{M}(X')}{\mathbb{E}} \left[ e^{(1-\alpha)(\xi+\rho\alpha)} \left( \frac{p_{\mathcal{M}(X)}(y,\rho)}{p_{\mathcal{M}(X')}(y,\rho)} \right)^\alpha \right] = \underset{(y,\epsilon) \sim \mathcal{M}_{\text{RDP}}(X')}{\mathbb{E}} \left[ e^{(1-\alpha)\epsilon} \left( \frac{p_{\mathcal{M}_{\text{RDP}}(X)}(y,\epsilon)}{p_{\mathcal{M}_{\text{RDP}}(X')}(y,\epsilon)} \right)^\alpha \right]. \tag{75}$$

The claimed equivalence follows immediately. $\qquad\square$

Lemma D.3 implies that variants of Theorems 4.2 to 4.6 hold for ex-post zCDP. In the variants of Theorems 4.4 to 4.6, the $\xi$ values of each involved mechanism must also be summed.

Now (71) implies the plain Gaussian + Brownian mechanism combination is $(\xi = 0)$-ex-post zCDP when returning $\rho = \frac{\epsilon + \epsilon_{\text{query}}}{\alpha}$. The SVT + Brownian mechanism combination is $(\xi = 9.292 \cdot 10^{-3})$-ex-post zCDP when returning $\rho = 3.541 \cdot 10^{-5} + \epsilon/\alpha$ by (72).

# E. Accuracy-First Differentially Private Optimization

In this section, we implement a variant of differentially private stochastic gradient descent (DP-SGD; Rajkumar & Agarwal, 2012; Song et al., 2013; Abadi et al., 2016), specifically DP-Adam, in the accuracy-first setting. We apply the algorithm to fine-tune an image classifier. After an initial fine-tuning run, we check the validation accuracy of the classifier, and then fine-tune longer if the accuracy is not high enough. As every iteration of fine-tuning is RDP, Theorem 4.5 gives an ex-post bound for a data-dependent number of iterations with public validation data, and Theorem 4.6 gives an ex-post bound with private validation data.

To compute RDP bounds for each iteration of fine-tuning, we use the Opacus library (Yousefpour et al., 2021). We use the same add/remove neighbourhood as in the synthetic data experiment (Section 6). We convert ex-post RDP bounds to ex-post ADP with the conversion theorem of Ghazi et al. (2025). For this conversion, Opacus computes ex-post RDP bounds for a set of $\alpha$ values, finds the $\epsilon_{\mathrm{adp}}$ at a given $\delta$ corresponding to each RDP bound, and takes the minimum $\epsilon_{\mathrm{adp}}$. With public validation data, each ex-post RDP bound is valid by Theorem 4.5, and each converted ex-post ADP bound is valid by the conversion theorem of Ghazi et al. (2025), so the minimum $\epsilon_{\mathrm{adp}}$ is valid.

Note that by default Opacus uses another conversion theorem (Balle et al., 2020) that is valid in the ex-ante setting and gives smaller ADP bounds. Future work should investigate if this conversion also holds in the ex-post setting.

With a private validation set, we use the plain Gaussian mechanism to check the accuracy threshold. The privacy accounting with this mechanism has five steps:

1. Find the optimal $(\alpha^*, \epsilon_{\mathrm{rdp}}^*)$ pair that gives the smallest $\epsilon_{\mathrm{adp}}$ for the initial optimisation run.

2. Set the accuracy check noise variance so that the composition of all checks is $(\alpha^*, \epsilon_{\mathrm{rdp}}^*)$-RDP.

3. Compute the $(\alpha^*, \epsilon_{\mathrm{rdp}}^*)$ pairs of the accuracy check for all other values of $\alpha$ in the set used for privacy accounting.

4. Take the maximum of the optimisation and accuracy check $\epsilon_{\mathrm{rdp}}$:s for each $\alpha$.

5. Convert those to ADP with the conversion of Ghazi et al. (2025).

Each ex-post RDP bound is valid by Theorem 4.6, so the conversion to ex-post ADP is valid.

As the specific setting, we fine-tune the last layer of a vision transformer (ViT-B-16, pre-trained on ImageNet-21K; Dosovitskiy et al., 2021) on CIFAR-10 (Krizhevsky, 2009), replicating an experiment from (Tobaben et al., 2023). To make the task harder, we only use 50 datapoints per class for training data, and another 50 datapoints per class for validation data. We use the standard CIFAR-10 test set. We set the initial $\epsilon_{\mathrm{adp}} = 1.5$, and use the same $\delta = 1/n_{\mathrm{train}}$ as Tobaben et al. (2023). Our initial $\epsilon_{\mathrm{adp}}$ is larger than the smallest one ($\epsilon_{\mathrm{adp}} = 1$) used by Tobaben et al. (2023) to compensate for the weaker conversion theorem of Ghazi et al. (2025). We use Adam (Kingma & Ba, 2015) as the optimiser, implemented with DP in Opacus (Yousefpour et al., 2021).

We tune hyperparameters for $\epsilon = 1.5$ using Optuna (Akiba et al., 2019), running 20 trials. We use the best hyperparameters for the rest of the experiment, except we divide the learning rate by 10 after the initial run of fine-tuning, since we expect the algorithm to have gotten reasonably close to a minimum at that point. We ignore the privacy cost of hyperparameter tuning, as is common in DP deep learning literature (Tobaben et al., 2023). Note that Tobaben et al. (2023) tune hyperparameters separately for every value of $\epsilon$, so we do not reach the same accuracy they do with other values of $\epsilon$.

With public validation data, we check the accuracy threshold every 50 epochs, and allow a maximum of 10 releases, including the initial one. With private validation data, we check accuracy every 100 epochs and allow 5 releases.

Results are plotted in Figures 5 and 6. They show that our ex-post private fine-tuning is able to reach the accuracy threshold while minimising $\epsilon_{\mathrm{adp}}$, with both public and private validation data. The threshold check has noise with private validation data, meaning that the accepted validation accuracy is sometimes below the threshold, but the distributions of final test accuracies are similar with both public and private validation data.

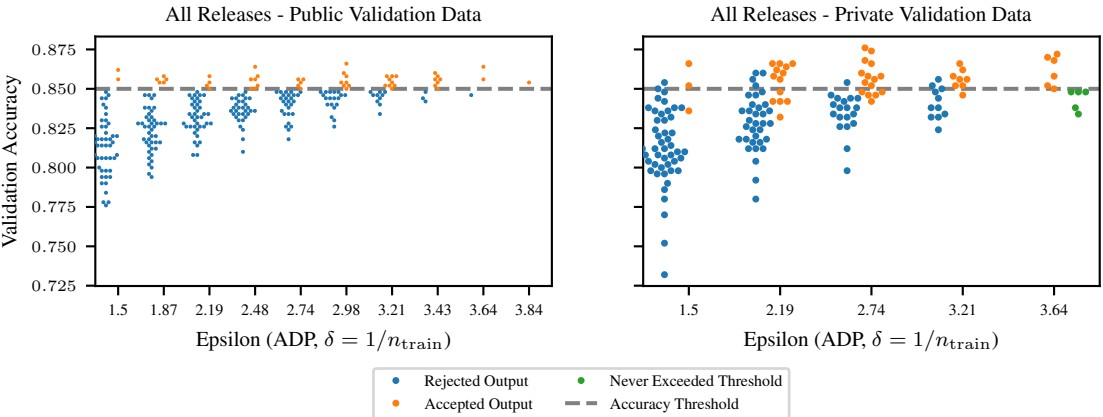

*Figure 5.* Accuracies and $\epsilon$ values for last-layer fine-tuning on CIFAR-10 with 50 examples per class, showing that ex-post DP-SGD is able to minimise the privacy cost while ensuring high accuracy, with both public (left) or private (right) validation data. The plot shows results from 50 repeats. The x-axis values are logarithmically spaced.

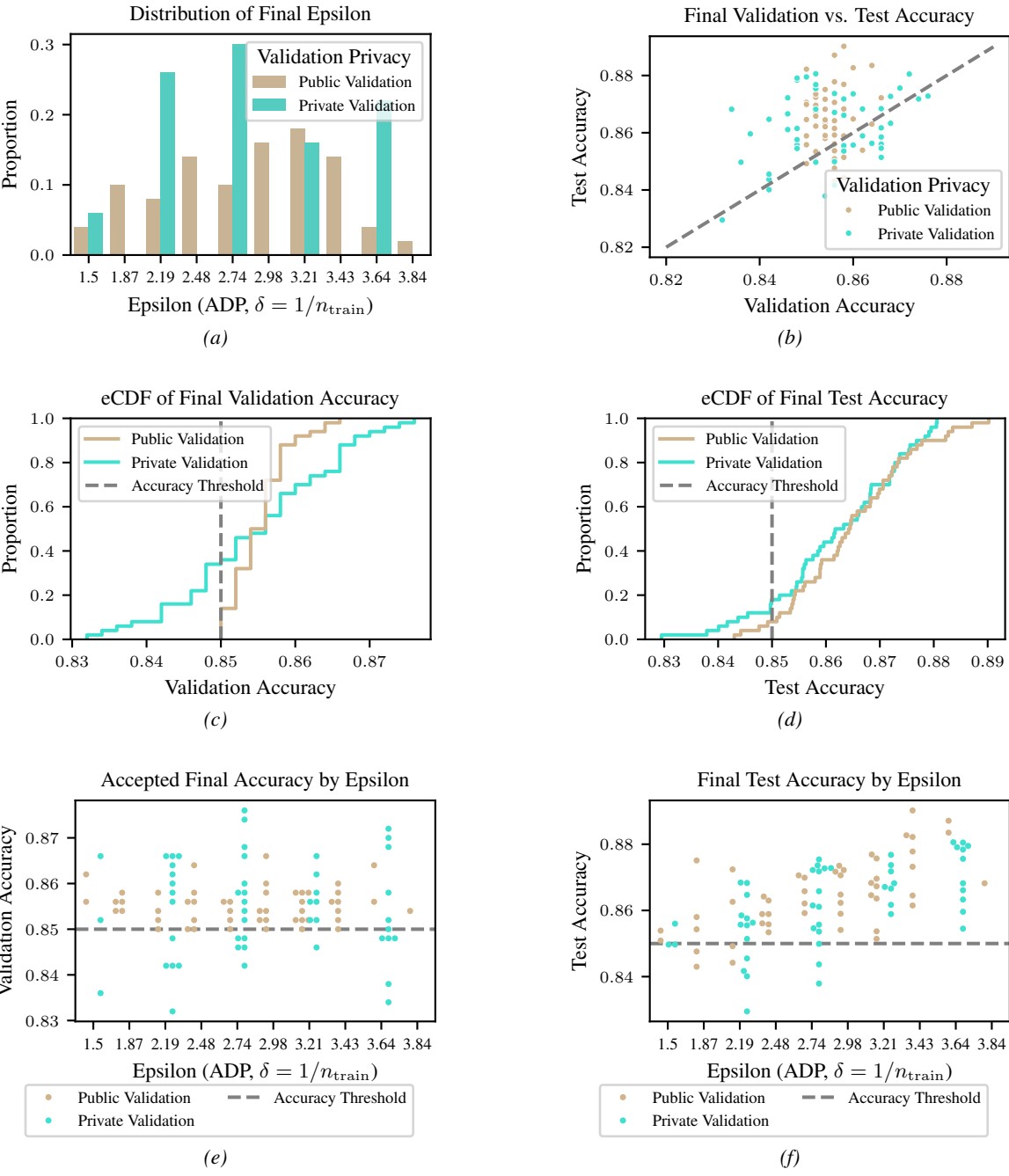

*Figure 6.* (a) Distribution of final $\epsilon$ values. (b) Scatterplot of validation and test accuracies of final results. There is no indication of overfitting to the validation set, as the points cluster around the diagonal gray line. (c, d) Empirical CDFs of final validation (c) and test (d) accuracies. The distribution of the validation accuracy is wider with a private validation set due to the accuracy check noise, but the distributions of test accuracy are similar. (e, f) Validation (e) and test (f) accuracies by $\epsilon$.

