# OpenReview forum: "Accuracy-First Rényi Differential Privacy and Post-Processing Immunity"
_ICML.cc/2026/Conference — ICML 2026 regular_

### Official Review · Reviewer_DP73 · 2026-02-24

**Soundness:** 4
**Presentation:** 4
**Significance:** 3
**Originality:** 3
**Overall Recommendation:** 5
**Confidence:** 3

**Summary:**

This paper investigates post-processing immunity of several accuracy first privacy definitions. Interestingly, this is largely ignored by past works. They first redefine some of the notation, which makes it easier to study post-processing. They then introduce ex-post RDP, which is the accuracy first analogue of Renyi DP. They prove various results, including post-processing immunity and study a Brownian mechanism for this definition of privacy.

**Compliance With Llm Reviewing Policy:**

Affirmed.

**Key Questions For Authors:**

I don't have any questions.

**Limitations:**

yes

**Strengths And Weaknesses:**

The paper is very well written and easy to follow. The problem is interesting, and to my knowledge, unexplored. This work advances our understanding of accuracy first DP, and provides a practical tool for generating such statistics. I don't see any issues with soundness or significance. I think it should be accepted.

---

> ### Author Rebuttal · Authors · 2026-03-30
>
> Thank you for the review. Please let us know if you have any suggestions for improvements.

---

> > ### Author Rebuttal · Reviewer_DP73 · 2026-03-31
> >
> > I have no concerns

---

### Official Review · Reviewer_6hr7 · 2026-02-26

**Soundness:** 4
**Presentation:** 3
**Significance:** 4
**Originality:** 4
**Overall Recommendation:** 5
**Confidence:** 4

**Summary:**

The author of the paper gives a broad context of an accuracy-first perspective on differential privacy (DP), where the privacy parameter is adaptively selected to satisfy an accuracy constraint rather than fixed ex ante. This submission considers the problem of working with ex-post (accuracy-first) privacy definitions rather than post-processing immunity, a property that is fundamental in standard DP but has not been formally established in prior ex-post frameworks.The author first introduces a new notation for ex-post privacy where the privacy bound is part of the mechanism’s output, enabling a coherent definition of post-processing. It then analyzes which existing ex-post definitions satisfy post-processing immunity. The authors show that pure ex-post privacy has post-processing immunity, while the probabilistic relaxation does not. Motivated by this, they propose an ex-post Rényi differential privacy (RDP) definition that enjoys post-processing immunity and supports mechanisms such as the Brownian mechanism. The paper develops core theoretical properties (post-processing, adaptive composition, conversion to ex-ante RDP under bounded ε) and demonstrates practicality through experiments on synthetic tabular data generation and image classifier fine-tuning.

**Compliance With Llm Reviewing Policy:**

Affirmed.

**Key Questions For Authors:**

### Key Questions for Authors

1. The post-processing immunity of $\\delta$-approximate ex-post privacy is marked as unknown. Do the authors have partial results or conjectures regarding necessary or sufficient conditions for immunity in that setting? Clarifying this could help complete the conceptual comparison across ex-post definitions.

2. In practice, how sensitive are the empirical results (Section 6) to the choice of $\\alpha$ in ex-post RDP? Would smaller values of $\\alpha$ significantly alter the privacy–utility tradeoffs or the stability of the adaptive procedure?

3. Can the authors clarify whether their ex-post RDP definition admits a clean interpretation analogous to $f$-DP or GDP, for example in terms of hypothesis testing or divergence geometry? Such an interpretation might improve intuition and broaden accessibility.

4. The Brownian mechanism equivalence (Theorem 5.2) is central to the paper. Could the authors elaborate on whether this alternative formulation meaningfully simplifies privacy accounting compared to stochastic-process-based analyses, particularly in practical implementations?

**Limitations:**

Yes. The authors explicitly discuss interpretability limitations of RDP and the lossy conversion to approximate DP, as well as broader interpretational challenges of ex-post bounds (Section 7 and Impact Statement). The discussion is thoughtful and adequate.

**Strengths And Weaknesses:**

## Strengths and Weaknesses



### Soundness

**Strengths**

- The theoretical development is generally rigorous and well-structured. The transition from classical RDP (Definition 2.2) to ex-post RDP (Definition 4.1) is conceptually coherent and technically consistent.

- The result that pure ex-post privacy satisfies post-processing immunity (Theorem 3.3), while probabilistic ex-post privacy does not (Theorem 3.5), is a useful clarification of the theoretical landscape.

- The adaptive composition theorems (Theorems 4.4–4.6) meaningfully extend ex-post privacy toward practical usage and demonstrate internal consistency of the proposed framework.

- The equivalence result for the Brownian mechanism (Theorem 5.2 and Corollary 5.3) is technically non-trivial and helps connect the proposal to known mechanisms in RDP.

**Weaknesses**

- Several key proofs rely heavily on appendices or prior work, and the main text sometimes lacks sufficient intuition to guide readers through the technical implications, particularly in Section 4. While this is acceptable for completeness, it reduces accessibility.

- The discussion around approximate ex-post privacy (Definition 2.5) and its post-processing immunity status remains incomplete ("?'' in Table 1). Although acknowledged, this leaves a noticeable gap in the conceptual comparison across definitions.

- Some claims (e.g., around practicality and robustness) would benefit from more explicit boundary conditions or counterexamples to clarify when the framework may fail or become overly conservative.

- The experimental section (Section 6) is relatively limited in scope compared to the theoretical contribution. The synthetic data experiment and fine-tuning example demonstrate feasibility but do not thoroughly stress-test the framework under diverse regimes or larger-scale benchmarks.

Overall, the work appears technically sound, but parts of the presentation and empirical validation could be strengthened to fully support the breadth of the claims.



### Presentation

**Strengths**

- The overall structure is logical: background → notation reformulation → post-processing analysis → new definition → properties → mechanism analysis → experiments.

- The introduction clearly motivates why post-processing immunity matters in ex-post settings and situates the work within ongoing discussions about accuracy-first privacy.

- Table 1 provides a helpful high-level comparison of different ex-post definitions and their properties.

- The limitations and impact statement are balanced and appropriately cautious.

**Weaknesses**

- The notation shift in Section 3.1 is subtle and may be difficult for readers not already familiar with ex-post privacy formulations. More explanatory text or examples would improve clarity.

- Transitions between key definitions (e.g., Definitions 3.1 and 4.1) could benefit from additional intuitive discussion explaining what conceptually changes and why it matters.

- The link between theoretical results and the experimental section is somewhat under-emphasized; it would help to explicitly tie specific empirical findings to specific theorems or properties.


### Significance

The paper addresses an important and timely issue. Accuracy-first privacy is increasingly relevant for real-world workflows where privacy budgets are not predetermined but adapted to accuracy requirements. However, without post-processing immunity, such frameworks risk being conceptually weaker than classical DP.

This work clarifies which existing ex-post definitions preserve post-processing immunity and proposes a new formulation that retains this property while supporting practical mechanisms. This strengthens the theoretical foundation of adaptive privacy accounting and could influence subsequent work in privacy accounting and mechanism design.

That said, the primary contribution is conceptual and theoretical; the empirical section mainly demonstrates feasibility rather than establishing broad practical impact.



### Originality

To the best of my knowledge, this work does not merely repackage known RDP results. In particular, it:

- Introduces a structural reformulation of ex-post mechanisms by embedding $\\epsilon$ in the output.

- Systematically analyzes post-processing immunity across existing ex-post definitions.

- Formally establishes the relationship between the Brownian mechanism and ex-post RDP.

- Develops composition results tailored to accuracy-first workflows.

Although ex-post RDP has also been proposed in parallel (Chua, Lynn, et al. "Bridging the Privacy Accounting Gap in DP-SGD." *Journal of Privacy and Confidentiality* 15.3 (2025).), the focus here on post-processing immunity and the structural reformulation appears to provide additional conceptual clarity.

---

> ### Author Rebuttal · Authors · 2026-03-30
>
> Thank you for the review. We have responded to each of your points below.
>
> > Several key proofs rely heavily on appendices or prior work, and the main text sometimes lacks sufficient intuition to guide readers through the technical implications, particularly in Section 4. While this is acceptable for completeness, it reduces accessibility.
>
> We will add proof ideas to each of the theorems in Section 4 in the revision.
>
> > The discussion around approximate ex-post privacy (Definition 2.5) and its post-processing immunity status remains incomplete ("?'' in Table 1). Although acknowledged, this leaves a noticeable gap in the conceptual comparison across definitions.
>
> > The post-processing immunity of $\delta$-approximate ex-post privacy is marked as unknown. Do the authors have partial results or conjectures regarding necessary or sufficient conditions for immunity in that setting? Clarifying this could help complete the conceptual comparison across ex-post definitions.
>
> We conjecture that post-processing immunity holds for $\delta$-approximate ex-post privacy since it holds for the ex-ante version. We agree that confirming or refuting this is a significant open question, but leave the investigation to future work due to our focus on ex-post RDP.
>
> > Some claims (e.g., around practicality and robustness) would benefit from more explicit boundary conditions or counterexamples to clarify when the framework may fail or become overly conservative.
>
> Can you clarify which specific claims you are referring to?
>
> > The experimental section (Section 6) is relatively limited in scope compared to the theoretical contribution. The synthetic data experiment and fine-tuning example demonstrate feasibility but do not thoroughly stress-test the framework under diverse regimes or larger-scale benchmarks.
>
> This is true, but the focus of the paper is on theory, so we think it is reasonable to leave large-scale applications of it to future work.
>
> > The notation shift in Section 3.1 is subtle and may be difficult for readers not already familiar with ex-post privacy formulations. More explanatory text or examples would improve clarity.
>
> We will add additional clarifications to Section 3.1 in the revision, including an example of the notation with the Brownian mechanism.
>
> > Transitions between key definitions (e.g., Definitions 3.1 and 4.1) could benefit from additional intuitive discussion explaining what conceptually changes and why it matters.
>
> We will add a paragraph discussing the difference between Definitions 3.1 and 4.1 in the revision.
>
> > The link between theoretical results and the experimental section is somewhat under-emphasized; it would help to explicitly tie specific empirical findings to specific theorems or properties.
>
> We will add additional references to relevant theorems in Section 6 in the revision.
>
> > In practice, how sensitive are the empirical results (Section 6) to the choice of $\alpha$ in ex-post RDP? Would smaller values of $\alpha$ significantly alter the privacy–utility tradeoffs or the stability of the adaptive procedure?
>
> We discuss the effects of changing $\alpha$ in Appendix D. Changing $\alpha$ changes $\epsilon$ according to mechanism-specific equations. The effect on the empirical results would be a shift of the $\epsilon$-axes in our figures.
>
> > Can the authors clarify whether their ex-post RDP definition admits a clean interpretation analogous to $f$-DP or GDP, for example in terms of hypothesis testing or divergence geometry? Such an interpretation might improve intuition and broaden accessibility.
>
> As discussed in Section 7, RDP and ex-post RDP do not have hypothesis testing interpretations like $f$-DP or GDP. We agree that the hypothesis testing interpretation is cleaner than the one for RDP, which is why we list developing ex-post versions of $f$-DP or GDP as important directions for future work.
>
> > The Brownian mechanism equivalence (Theorem 5.2) is central to the paper. Could the authors elaborate on whether this alternative formulation meaningfully simplifies privacy accounting compared to stochastic-process-based analyses, particularly in practical implementations?
>
> The alternative formulation does simplify privacy accounting, since it can be based on Theorems 4.2 and 4.5 with a short proof (in Appendix C.4).

---

> > ### Author Rebuttal · Reviewer_6hr7 · 2026-04-03
> >
> > Thank you for the detailed rebuttal and clarifications. Several of my concerns have been addressed at a high level, particularly regarding planned improvements to exposition (e.g., adding proof sketches and additional intuition in Section 4). These revisions should help improve readability and accessibility.
> >
> > However, some concerns remain only partially resolved:
> >
> > - The conceptual gap around approximate ex-post privacy persists. While I understand the decision to leave this as future work, the current discussion remains somewhat limited. Providing even partial insights (e.g., conditions, intuition, or barriers) would strengthen the comparison across definitions.
> >
> > -The connection between theory and experiments is still somewhat underdeveloped. While the authors indicate that they will improve this, the rebuttal does not yet clearly articulate how specific empirical results validate or illustrate particular theoretical properties.
> >
> > - The claim that the alternative formulation of the Brownian mechanism simplifies privacy accounting is promising, but the explanation remains relatively high-level. A more concrete illustration or example would improve clarity and practical understanding.
> >
> > Follow-up question:
> > Can the authors provide a more explicit characterization (even informally) of scenarios where ex-post RDP may become overly conservative or harder to interpret in practice, particularly when adaptive ε values vary significantly across outputs?
> >
> > Overall, while the rebuttal is helpful and constructive, some points would benefit from deeper clarification in the final version.

---

> > > ### Author Response · Authors · 2026-04-07
> > >
> > > Thank you for the reply. We have responded to your comments and question below.
> > >
> > > > The conceptual gap around approximate ex-post privacy persists. While I understand the decision to leave this as future work, the current discussion remains somewhat limited. Providing even partial insights (e.g., conditions, intuition, or barriers) would strengthen the comparison across definitions.
> > >
> > > > The connection between theory and experiments is still somewhat underdeveloped. While the authors indicate that they will improve this, the rebuttal does not yet clearly articulate how specific empirical results validate or illustrate particular theoretical properties.
> > >
> > > The goal of the experiments is to showcase how the theory overall can be used in applications, not to validate individual theorems. The synthetic data experiment in Section 6 illustrates Theorems 4.4 and 4.6, as well as Corollary 5.3. The image classification experiment in Appendix E shows how Theorem 4.5 can be applied to DP-SGD, and also compares the difference of public and private validation data, corresponding to Theorems 4.5 and 4.6, respectively.
> > >
> > > > The claim that the alternative formulation of the Brownian mechanism simplifies privacy accounting is promising, but the explanation remains relatively high-level. A more concrete illustration or example would improve clarity and practical understanding.
> > >
> > > The alternative formulation allows applying generic theorems, specifically Theorems 4.2 and 4.5, which is why the proof is so short. In contrast, analysing the Brownian mechanism directly would require carefully analysing the dependencies between the released values and privacy bounds, which would require duplicating much of the work in the proof of Theorem 4.5.
> > >
> > > > Follow-up question: Can the authors provide a more explicit characterization (even informally) of scenarios where ex-post RDP may become overly conservative or harder to interpret in practice, particularly when adaptive ε values vary significantly across outputs?
> > >
> > > The DP-SGD experiment in Appendix E is likely a good example. The privacy random variable (PRV)-accountant of Opacus returns smaller $\epsilon$ values than the RDP accountant (Tobaben et al. 2023). On top of that, as discussed in Appendix E, we use a less efficient conversion theorem than the RDP accountant uses, since an ex-post equivalent of that conversion theorem has not been proven. As a result, there is likely room to improve the privacy bounds of the experiment in future work.
> > >
> > > The interpretation differences between RDP and $(\epsilon, \delta)$-DP or $f$-DP apply in any scenario, for both ex-ante and ex-post cases. Significantly varying $\epsilon$ values are inherently challenging to interpret in ex-post settings, regardless of the specific definition that is used.

---

### Official Review · Reviewer_Hffw · 2026-03-05

**Soundness:** 3
**Presentation:** 2
**Significance:** 3
**Originality:** 3
**Overall Recommendation:** 4
**Confidence:** 2

**Summary:**

This paper proposes a novel accuracy-first Rényi differential privacy framework in which the privacy bound is determined in a data-dependent manner while preserving post-processing immunity, enabling adaptive privacy calibration with practical mechanisms such as the Brownian mechanism.

**Compliance With Llm Reviewing Policy:**

Affirmed.

**Key Questions For Authors:**

None

**Limitations:**

yes

**Strengths And Weaknesses:**

Strengths:
1. The paper identifies a critical limitation in existing ex-post differential privacy definitions, namely the lack of post-processing immunity, and introduces a new formalism that enables a rigorous and complete analysis of this property.
2. The work proposes a theoretically well-founded α-ex-post RDP definition and successfully incorporates the Brownian mechanism into the framework, thereby bridging the gap between rigorous privacy theory and practical data analysis tools.
3. The paper presents a novel and practical early-stopping algorithm based on a private validation set, effectively overcoming the limitation of prior approaches that assess accuracy only on the training data.
Weakness:
1.The paper lacks a discussion of worst-case tail bounds. Privacy loss is controlled only in expectation, which does not allow intuitive characterization of the probability of extreme privacy breaches in a single run, as $(\epsilon, \delta)$-DP does.
2.Algorithm 1 relies on a private validation set to trigger stopping, which can reduce training utility when data is limited, and the paper does not analyze the trade-off between this utility loss and the privacy gain from early stopping.
3.The some notation error in the paper. For example, in Theorem 4.4 and Theorem 4.5, the symbols for the iteration upper bound are inconsistently used ($T$ versus $K$). In addition, the first line of Algorithm 1 contains $\epsilon_{i:1-1}$.

---

> ### Author Rebuttal · Authors · 2026-03-30
>
> Thank you for the review. We have responded to your points below.
>
> > Weakness: 1.The paper lacks a discussion of worst-case tail bounds. Privacy loss is controlled only in expectation, which does not allow intuitive characterization of the probability of extreme privacy breaches in a single run, as $(\epsilon, \delta)$-DP does.
>
> Ex-post RDP can be converted to ex-post approximate DPADP (Definition 2.5 in the paper) by Lemma 7 from Ghazi et al. (2025), which has the same characterisation of the probability of extreme privacy breaches as $(\epsilon, \delta)$-DP.
>
> > 2.Algorithm 1 relies on a private validation set to trigger stopping, which can reduce training utility when data is limited, and the paper does not analyze the trade-off between this utility loss and the privacy gain from early stopping.
>
> We ran an additional experiment where we varied the proportions allocated to the validation and training datasets in the setting of the synthetic data experiment in Section 6. Results and details are here: https://anonymous.4open.science/r/accuracy-first-rebuttal-skfh354rff/rebuttal-plots.pdf. The results show that there is a trade-off where smaller validation sets and larger training sets lead to better test accuracy, but at the cost of a larger final $\epsilon$ value. We will add these results to the revised paper.
>
> > 3.The some notation error in the paper. For example, in Theorem 4.4 and Theorem 4.5, the symbols for the iteration upper bound are inconsistently used ($T$ versus $K$). In addition, the first line of Algorithm 1 contains $\epsilon_{i:1-1}$.
>
> Using $T$ in Theorem 4.4 and $K$ in Theorem 4.5 is intentional, since the $K$ in Theorem 4.5 denotes the arbitrary maximum number of mechanisms that can be run, not how many are actually run. We will clarify this in the revision. Thank you for pointing out the other issue, we will fix it in the revision.

---

> > ### Author Rebuttal · Reviewer_Hffw · 2026-04-03
> >
> > Thank you for the response. The authors have addressed my concerns.

---

> > > ### Author Response · Authors · 2026-04-07
> > >
> > > Thank you for confirming we addressed your concerns. As a result, would you consider increasing the score?

---

### Official Review · Reviewer_VTN1 · 2026-03-14

**Soundness:** 3
**Presentation:** 3
**Significance:** 2
**Originality:** 3
**Overall Recommendation:** 4
**Confidence:** 3

**Summary:**

This paper addresses a critical gap in ex-post differential privacy (DP) by proposing a new notation that incorporates the privacy bound as part of the algorithm's output, enabling  post-processing immunity proofs. The authors investigate existing ex-post definitions, demonstrating that pure ex-post privacy enjoys post-processing immunity while probabilistic ex-post privacy does not. Also they introduce ex-post Rényi DP (RDP), prove its fundamental properties, and establish that the Brownian mechanism satisfies ex-post RDP—the first such proof for a definition with post-processing immunity. Practical experiments on synthetic data generation and private fine-tuning validate the framework's utility.

**Compliance With Llm Reviewing Policy:**

Affirmed.

**Key Questions For Authors:**

Could you please give a detailed explanation of the statement "However, we have not been able to come up with a satisfactory construction of  Epsilon' that works for all f." (left column Page 4)? The argument after the claim looks confusing.

**Limitations:**

yes

**Strengths And Weaknesses:**

Strengths:

 (1) The authors provide Comprehensive proofs which establish ex-post RDP's desirable properties.

(2) The paper is well-motivated and mathematically sound.

Weakness:

 (1) My main concern is about Definitions 3.1 and 3.2, which incorporate the privacy bound as part of the algorithm's output, which make these two definitions quite restricted. In previous works ((Ligett et al. 2017; Whitehouse et al. 2022), the privacy mechanisms and the data-dependent privacy bound Epsilon are independent of each other, which is the case for real-world situations. However, since the algorithm output in Definition 3.1 includes this privacy bound, the privacy algorithm and the privacy bound are not independent, which deviates from the practical applications.

 (2) The postprocessing function f should be quite arbitrary. However, the postprocessing function in Definition 3.2 is restricted. So the ex-post privacy mechanisms are just special cases of those in the above previous works ((Ligett et al. 2017; Whitehouse et al. 2022).

 (3) Minor: A confusing statement: “We do not necessarily assume that P is continuous, but rather assume that P has a density function with regards to some base measure”. If P has a density function, then it must be continuous.

---

> ### Author Rebuttal · Authors · 2026-03-30
>
> Thank you for the review. We have responded to each of your points and the question below.
>
> > (1) My main concern is about Definitions 3.1 and 3.2, which incorporate the privacy bound as part of the algorithm's output, which make these two definitions quite restricted. In previous works ((Ligett et al. 2017; Whitehouse et al. 2022), the privacy mechanisms and the data-dependent privacy bound Epsilon are independent of each other, which is the case for real-world situations. However, since the algorithm output in Definition 3.1 includes this privacy bound, the privacy algorithm and the privacy bound are not independent, which deviates from the practical applications.
>
> Incorporating the privacy bound as an output is only a conceptual change from the previous works, and does not restrict the practical applicability of the definitions. As discussed in the last paragraph of Section 3.1, it is possible to fit an independent privacy bound into our definitions by making the algorithm output the value that the $\mathcal{E}$ function from the previous works returns.
>
> > (2) The postprocessing function f should be quite arbitrary. However, the postprocessing function in Definition 3.2 is restricted. So the ex-post privacy mechanisms are just special cases of those in the above previous works ((Ligett et al. 2017; Whitehouse et al. 2022).
>
> The only restriction we place on the post-processing is that it cannot change the claimed privacy bound of the algorithm, but otherwise it can be completely arbitrary. This restriction is in line with the capabilities of a realistic adversary, who cannot change the privacy bound that the party that originally ran the algorithm claimed. Besides, without this restriction, no definition would have post-processing immunity, since the post-processing could just set the returned privacy bound to 0. We will clarify this in the revision.
>
> > (3) Minor: A confusing statement: “We do not necessarily assume that P is continuous, but rather assume that P has a density function with regards to some base measure”. If P has a density function, then it must be continuous.
>
> The statement “P has a density function with regards to some base measure” means that there is a measure $\mu$ and a function $p_P$ such that for any measurable set $A$, $P(A) = \int_A p_P(y) \mathrm{d} \mu(y)$. $p_P$ is then the density function with regards to $\mu$. For continuous $P$, $\mu$ is the Lebesgue measure, but density functions with regards to other measures can exist for non-continuous random variables. For example, the probability mass function of a discrete random variable is a density function with regards to the counting measure. We will add the precise definition in the revision.
>
> > Could you please give a detailed explanation of the statement "However, we have not been able to come up with a satisfactory construction of Epsilon' that works for all f." (left column Page 4)? The argument after the claim looks confusing.
>
> The preceding paragraph shows that stating post-processing immunity for the ex-post definitions in previous works would require constructing a new privacy bound function $\mathcal{E}’$ for the post-processed algorithm that is equivalent to the original $\mathcal{E}$. The argument afterwards shows that one candidate construction that at first seems reasonable does not in fact work. This is because the candidate would output extremely pessimistic privacy bounds in the case we present in the argument, so it is not equivalent to the original privacy bound function $\mathcal{E}$. We will clarify this in the revision.

---

> > ### Author Rebuttal · Reviewer_VTN1 · 2026-04-08
> >
> > (*) For my third question about the continuity of P,  P must be continous if it has a density function. So the rebuttal is not correct.
> >
> > (**) I don't think that this restriction that the adversary cannot change the privacy bound that the party that originally ran the algorithm claimedis is in line with the capabilities of a realistic adversary. In reality, when we add calibrated privacy noises,  we reduce this privacy bound indeed.

---

### Decision · Program_Chairs · 2026-04-30

**Decision:**

Accept (regular)

**Comment:**

The paper studies theoretical aspects of the "accuracy-first perspective" on privacy, particularly, on its robustness to postprocessing. They develop a new framework for this and study existing methods under it. The reviewers are all aligned on the value of the theory and general presentation quality of the submission. I support acceptance of the paper.